# Nonlinearly Preconditioned Gradient Methods: Momentum and Stochastic Analysis

**Konstantinos Oikonomidis**[*]
ESAT-STADIUS & Leuven.AI
KU Leuven
konstantinos.oikonomidis@kuleuven.be

**Jan Quan**[*]
ESAT-STADIUS & Leuven.AI
KU Leuven
jan.quan@kuleuven.be

**Panagiotis Patrinos**
ESAT-STADIUS & Leuven.AI
KU Leuven
panos.patrinos@kuleuven.be

## Abstract

We study nonlinearly preconditioned gradient methods for smooth nonconvex optimization problems, focusing on sigmoid preconditioners that inherently perform a form of gradient clipping akin to the widely used gradient clipping technique. Building upon this idea, we introduce a novel heavy ball-type algorithm and provide convergence guarantees under a generalized smoothness condition that is less restrictive than traditional Lipschitz smoothness, thus covering a broader class of functions. Additionally, we develop a stochastic variant of the base method and study its convergence properties under different noise assumptions. We compare the proposed algorithms with baseline methods on diverse tasks from machine learning including neural network training.

## 1 Introduction and preliminaries

In this paper we consider general minimization problems of the form:

$$\min_{x \in \mathbb{R}^n} f(x), \tag{P}$$

where $f : \mathbb{R}^n \to \mathbb{R}$ is a smooth and possibly nonconvex function. Under (global) Lipschitz continuity of the gradient, gradient descent (GD) and its stochastic counterpart, stochastic gradient descent (SGD) are the standard methods to efficiently tackle such problems. Gradient descent-type methods are the backbone of training machine learning models, especially considering the ever-growing size of datasets and architectures. Nevertheless, most modern-day applications involve cost functions that do not fit into the traditional Lipschitz gradient assumption [48]. Although GD and SGD, being versatile solvers, can handle such problems as well, they require careful tuning or expensive linesearch strategies to converge. In recent years, there has thus been considerable effort to introduce and analyze methods that can better adapt to more general smoothness conditions.

Gradient clipping [48, 20] is a standard practice in tasks such as language models and has been widely used to stabilize the training of neural networks. Recently, the method has received a lot of theoretical attention, especially since it was shown to be effective for problems under a less restrictive notion of smoothness, called $(L_0, L_1)$-smoothness [48]. In [36], it was shown that in fact a whole family of clipping methods, including the soft clipping algorithm [47] can be viewed as simple gradient

---

[*]Equal contribution.

39th Conference on Neural Information Processing Systems (NeurIPS 2025).

Table 1: Examples of reference functions along with generated algorithms. The subscript $i$ denotes the coordinate-wise update. We call the method generated by $\cosh -1$ Hyperbolic Gradient Descent (HGD), while i indicates that the method is generated by an isotropic reference function and s by a separable one (see also Subsection 1.4).

| $\phi(x)$ | Update | Algorithm |
|---|---|---|
| $\varepsilon(-\|x\| - \ln(1 - \|x\|))$ | $x^{k+1} = x^k - \frac{\gamma}{\varepsilon + \|\nabla f(x^k)\|} \nabla f(x^k)$ | NGD [48, Equation (6)] |
| $\varepsilon \sum_{i=1}^n (-|x_i| - \ln(1 - |x_i|))$ | $x_i^{k+1} = x_i^k - \frac{\gamma}{\varepsilon + |\nabla_i f(x^k)|} \nabla_i f(x^k)$ | sNGD |
| $\cosh(\|x\|) - 1$ | $x^{k+1} = x^k - \gamma \frac{\operatorname{arsinh}(\|\nabla f(x^k)\|)}{\|\nabla f(x^k)\|} \nabla f(x^k)$ | iHGD |
| $\sum_{i=1}^n \cosh(x_i) - 1$ | $x_i^{k+1} = x_i^k - \gamma \operatorname{arsinh}(\nabla_i f(x^k))$ | sHGD |

descent with nonlinear preconditioning: given $x^0 \in \mathbb{R}^n$ and a stepsize $\gamma > 0$, update $x^k$ as

$$x^{k+1} = x^k - \gamma \nabla \phi^*(\nabla f(x^k)). \tag{1}$$

In this scheme, $\phi : \mathbb{R}^n \to \overline{\mathbb{R}}$ is a convex function that plays the role of a *reference function*, while its convex conjugate, $\phi^*$ is called the *dual reference function* and generates the preconditioner $\nabla \phi^*$. The fact that this method is connected to the gradient clipping framework can be seen by choosing a strongly convex reference function $\phi$ that has bounded domain. Then, the mapping $\nabla \phi^*$ maps $\mathbb{R}^n$ to the unit $n$-ball and as such, naturally clips the gradient. In fact, for $\phi(x) = \varepsilon(-\|x\| - \ln(1 - \|x\|))$ with $\varepsilon > 0$, (1) takes the form of the popular normalized gradient descent method (NGD), as displayed in the first line of Table 1. It is moreover important to stress the versatility of the framework, in that it also involves algorithms that normalize the gradient per component, akin to the methods that are used in practice. Such an example is presented in the second line of Table 1.

In fact, the method in (1) is naturally generated from a majorization-minimization perspective, where in each iteration the following nonlinear upper bound is minimized w.r.t. $x$:

$$f(x) \leq f(x^k) + \frac{1}{L}\phi(L(x - y^k)) - \frac{1}{L}\phi(L(x^k - y^k)), \tag{2}$$

where $y^k = x^k - \frac{1}{L}\nabla \phi^*(\nabla f(x^k))$. This inequality is called the anisotropic descent inequality [24] and is a natural extension of Lipschitz smoothness that is tightly connected to the notion of $\Phi$-convexity [1] and to optimal transport theory [45]. We present more details on the concept of $\Phi$-convexity in Appendix A.

Intuitively, in order to generate less restrictive descent inequalities, one would consider reference functions $\phi$ that grow rapidly, thus making the r.h.s. of (2) as large as possible. This procedure leads to algorithms similar to the one in the first line of Table 1. Nevertheless, one can follow a different procedure and generate a descent inequality that is a tighter model of the cost function, thus obtaining an algorithm with faster convergence. Consider for example $\phi(x) = \cosh(\|x\|) - 1$. This function grows faster than any polynomial in $\|x\|$ and is strongly convex, implying that the algorithm presented in the third line of Table 1 can handle problems beyond Lipschitz smoothness (from [36, Proposition 2.3]). However, in contrast to gradient clipping, the generated method does not clip the gradient to a predefined value, but rather rescales it adaptively, leading to an implicit adaptive stepsize rule. In fact, for reference functions that grow faster than $\frac{1}{2}\|x\|^2$, the descent inequality is less restrictive than the standard Euclidean descent, and the preconditioner naturally takes a sigmoid shape. For example, $\phi(x) = \cosh(\|x\|) - 1$ leads to $\nabla \phi^*(y) = \operatorname{arsinh}(\|y\|)\frac{y}{\|y\|}$.

Nevertheless, although considerable effort has been dedicated to analyzing the gradient clipping method, general methods of the form (1) are much less explored. In this paper we aim at bridging this gap, by introducing a heavy ball-type algorithm based on (1) and performing an initial analysis of a stochastic extension of (1).

## 1.1 Our contribution

We summarize the contributions of the paper in the following.

- We extend the anisotropic gradient descent method [32, 24, 36] by incorporating heavy-ball momentum in the base iterate and study the convergence guarantees of the proposed method

in the general nonconvex setting. To the best of our knowledge, such a scheme has not been analyzed before in the full generality of the setting considered in this paper. We remark that our analysis is based on the anisotropic descent inequality, a condition that is less restrictive than $(L_0, L_1)$-smoothness and is applicable to a whole family of algorithms. Moreover, we prove that our scheme enjoys a linear rate of convergence under a generalized PL condition.

- We propose a stochastic extension of the base method and analyze it under different noise assumptions. We first prove its approximate convergence under a condition that we show is less restrictive than bounded variance and then provide a convergence rate for interesting reference functions assuming bounded variance and unbiasedness of the stochastic oracle. We moreover study the method under a generalized PL inequality that leads to a linear rate up to a constant.

- In numerical simulations we show that the proposed methods perform well on a variety of machine learning problems, including neural network training and matrix factorization.

## 1.2 Related work

**Dual space preconditioning / anisotropic gradient descent.** The general scheme presented in (1) was originally introduced in [32] for convex, essentially smooth problems, where it was studied under a generalization of cocoercivity called dual relative smoothness. The anisotropic descent inequality, which lies at the center of our analysis was introduced in [26] and was further studied in [24], where also a proximal extension of the base method (1) was introduced in order to tackle nonconvex, composite nonsmooth minimization problems. Further results were then provided under a generalized convexity point of view in [28, 35], while a generalization of the method in measure spaces was presented in [4]. Moreover, a nonlinear proximal point method for monotone operators was analyzed in [25], while the regularity properties of the anisotropic proximal mapping where a general nonlinear strictly convex prox function is used, were studied in [27, 23]. Recently in [36], the method was shown to encompass a variety of popular algorithms including but not limited to the clipped and normalized gradient method. Finally in [18] an accelerated extension of (1) for Lipschitz smooth functions was introduced and studied.

**Gradient clipping and generalized smoothness.** Departing from the anisotropic gradient descent framework, the algorithms that we study are mostly connected to clipped and normalized gradient methods. Although the literature in this field is vast and ever-expanding we refer to some of the major works in the following. In [48] the notion of $(L_0, L_1)$-smoothness was introduced and the first theoretical analysis of gradient clipping under this new condition was presented. This approach was later extended in [47], where a general clipping framework with momentum under $(L_0, L_1)$-smoothness was analyzed in the deterministic and the stochastic setting. [8] studied an algorithm that used both clipping and normalization to tackle more general noise assumptions, further deepening the theoretical understanding of the method. A large body of work has since been devoted to studying such methods under relaxed noise and smoothness assumptions [17, 20, 7, 30, 15, 16, 29, 46]. Moreover, for convex problems, in [11], a stochastic clipping method where the clipping is performed using an independent sample was analyzed.

The notion of $(L_0, L_1)$-smoothness has also received widespread attention beyond the gradient clipping literature. In [44] new interesting methods for $(L_0, L_1)$-smooth problems that take the form of rescaled gradient descent were introduced and studied, and an accelerated variant for convex problems was proposed. An analysis for convex problems was performed in [12] where also a stochastic variant of the proposed method under interpolation was studied.

## 1.3 Notation

We denote by $\langle \cdot, \cdot \rangle$ the standard Euclidean inner product on $\mathbb{R}^n$ and by $\| \cdot \|$ the standard Euclidean norm on $\mathbb{R}^n$ as well as the spectral norm for matrices. We denote by $\mathcal{C}^k(Y)$ the class of functions which are $k$ times continuously differentiable on an open set $Y \subseteq \mathbb{R}^n$. For a proper function $f : \mathbb{R}^n \to \overline{\mathbb{R}}$ with $\overline{\mathbb{R}} := \mathbb{R} \cup \{+\infty\}$ and $\lambda \geq 0$ we define the episcaling $(\lambda \star f)(x) = \lambda f(\lambda^{-1} x)$ for $\lambda > 0$ and $(\lambda \star f)(x) = \delta_{\{0\}}(x)$ otherwise. For an $f \in \mathcal{C}^2(\mathbb{R}^n)$ we say that it is $(L_0, L_1)$-smooth for some $L_0, L_1 > 0$ if it holds that $\|\nabla^2 f(x)\| \leq L_0 + L_1 \|\nabla f(x)\|$ for all $x \in \mathbb{R}^n$. Otherwise we adopt the notation from [42].

## 1.4 The anisotropic descent inequality

Our analysis is centered around the anisotropic descent inequality from [24], that was recently shown to be connected to other notions of generalized smoothness in [36] and is itself a globalization of anisotropic prox-regularity [23, Definition 2.13].

**Definition 1.1** (anisotropic descent inequality). *Let $f \in \mathcal{C}^1(\mathbb{R}^n)$ such that the following constraint qualification holds true*

$$\operatorname{rge} \nabla f \subseteq \operatorname{rge} \nabla \phi. \tag{3}$$

*Then we say that $f$ satisfies the anisotropic descent property (is anisotropically smooth) relative to $\phi$ if for all $x, \bar{x} \in \mathbb{R}^n$*

$$f(x) \leq f(\bar{x}) + \tfrac{1}{L} \star \phi(x - \bar{y}) - \tfrac{1}{L} \star \phi(\bar{x} - \bar{y}), \tag{4}$$

*where $\bar{y} = \bar{x} - \frac{1}{L}\nabla\phi^*(\nabla f(\bar{x}))$.*

Although in [36] a different form of anisotropic smoothness that contains two constants $L, \bar{L} > 0$ was presented, w.l.o.g. we present the simpler version displayed above since the constant $\bar{L}$ can be absorbed into $\phi$ by considering $\tilde{\phi} := \bar{L}\phi$.

In this paper, we focus mostly on strongly convex reference functions that grow faster than the quadratic, in order to analyze the method under less restrictive conditions than Lipschitz smoothness via [36, Proposition 2.3]. Keeping in line with the related literature, for some kernel function $h : \mathbb{R} \to \overline{\mathbb{R}}$, we refer to functions $\phi = h \circ \| \cdot \|$ as *isotropic* and $\phi(x) = \sum_{i=1}^n h_i(x_i)$ as *anisotropic* or *separable*. The kernel function $h = \cosh -1$ has some interesting properties that we focus on throughout the paper: it is strongly convex, grows faster than any polynomial and has full domain, thus leading to a preconditioner that does not fully clip the gradient, but rather rescales it adaptively. We call the method generated by $\cosh -1$ the *hyperbolic gradient descent* (HGD) method.

We next formulate our assumptions on $\phi$, which we consider valid throughout the rest of the paper.

**Assumption 1.2.** *The reference function $\phi : \mathbb{R}^n \to \overline{\mathbb{R}}$ is strongly convex and even with $\phi(0) = 0$. $\operatorname{int} \operatorname{dom} \phi \neq \emptyset$; $\phi \in \mathcal{C}^2(\operatorname{int} \operatorname{dom} \phi)$ and for any sequence $\{x^k\}_{k \in \mathbb{N}_0}$ that converges to some boundary point of $\operatorname{int} \operatorname{dom} \phi$, $\|\nabla\phi(x^k)\| \to +\infty$.*

Note that under Assumption 1.2 $\phi^* \in \mathcal{C}^2(\mathbb{R}^n)$ from [41, p. 42], while $\nabla\phi^*$ is globally Lipschitz continuous and we can omit the constraint qualification in the definition of anisotropic smoothness. Moreover, from [2, Proposition 11.7], $\arg\min\phi = \{0\}$ and thus $\phi \geq 0$. Having formulated our assumption on the reference function, we now move on to the assumptions on the cost function.

**Assumption 1.3.** *The cost function $f : \mathbb{R}^n \to \mathbb{R}$ is anisotropically smooth with constant $L > 0$ and $f_\star = \inf f > -\infty$.*

It is important to note that under generally mild conditions on $\phi$, anisotropic smoothness follows from a second-order characterization [36, Lemma 2.5]:

$$\nabla^2 f(x) \prec L[\nabla^2\phi^*(\nabla f(x))]^{-1} \qquad \forall x \in \mathbb{R}^n, \tag{5}$$

which leads to a connection with the popular $(L_0, L_1)$-smoothness condition [48]:

**Remark 1.4** (connection between $(L_0, L_1)$- and anisotropic smoothness). In light of [36, Corollary 2.11], any $f \in \mathcal{C}^2(\mathbb{R}^n)$ that is $(L_0, L_1)$-smooth, is also anisotropically smooth relative to $\phi(x) = \frac{L_0}{L_1}(-\|x\| - \ln(1 - \|x\|))$ with any constant $L < L_1$. It is important to stress that this result is due to a simplification of the matrix inequality displayed above and as such, anisotropic smoothness actually holds with tighter constants in many interesting cases. In fact, as shown in [3, Section A.3], anisotropic smoothness for this specific reference function is *less restrictive* than $(L_0, L_1)$-smoothness, since there exist continuously differentiable functions that are not $(L_0, L_1)$-smooth but are anisotropically smooth.

We stress that our stationarity measure throughout the paper is $\phi(\nabla\phi^*(\nabla f(x)))$, which is the natural measure for this algorithmic family. When specifying $\phi$, one can in many cases also translate the results to the standard measure $\|\nabla f(x)\|$. For example, for $\phi = \cosh \circ \| \cdot \| - 1$, we have that $\phi(\nabla\phi^*(\nabla f(x))) = \sqrt{1 + \|\nabla f(x)\|^2} - 1$ and thus the analysis follows similarly to [36, Corollary 3.3].

## 2 The nonlinearly preconditioned gradient method with momentum

In this section we propose and analyze a novel heavy ball-type method based on (1). The proofs can be found in Appendix C.

---

**Algorithm 1** Nonlinearly preconditioned gradient method with momentum (m-NPGM)

---

**Require:** Choose $x^0 \in \mathbb{R}^n$, $\gamma, \beta > 0$ and set $m^{-1} = 0_n$.
**Repeat for** $k = 0, 1, \dots$ until convergence
 1: Compute
$$m^k = \beta m^{k-1} + (1 - \beta)\nabla\phi^*(\nabla f(x^k)). \tag{6}$$

 2: Compute
$$x^{k+1} = x^k - \gamma m^k. \tag{7}$$

---

**Remark 2.1.** Before moving on to our convergence results, we have to note that our proposed method Algorithm 1 is different from the methods in the related literature. Specifically, our momentum estimate consists of convex combinations of the *preconditioned gradients*, in contrast to the standard technique of aggregating the gradients and then preconditioning $m^k$ in the update of $x^{k+1}$ [16, 39, 40, 43]. With simple algebraic manipulations, the algorithm can be written equivalently as

$$x^{k+1} = x^k - (1 - \beta)\gamma\nabla\phi^*(\nabla f(x^k)) + \beta(x^k - x^{k-1}),$$

and thus takes the form of the standard heavy ball method but applied to the mapping $\nabla\phi^* \circ \nabla f$. We believe that this is a more natural approach for the descent inequality we base our analysis on. This is made more apparent with the introduction of our Assumption 2.5.

The following theorem describes the convergence of Algorithm 1 under anisotropic smoothness of $f$. To the best of our knowledge this is the first result regarding the convergence of the method with momentum.

**Theorem 2.2.** *Let Assumption 1.3 hold and $\{x^k\}_{k\in\mathbb{N}_0}$ be the sequence of iterates generated by Algorithm 1 with $\beta \in [0, 0.5)$ and $\gamma = \frac{\alpha}{L}$, $\alpha \leq 1$. Then, we have the following rate:*

$$\min_{0 \leq k \leq K} \phi(\nabla\phi^*(\nabla f(x^k))) \leq \frac{L(f(x^0) - f_\star)}{\alpha(K + 1)(1 - 2\beta)}. \tag{8}$$

The proof of Theorem 2.2 requires some additional effort compared to standard heavy ball momentum analysis under Lipschitz or $(L_0, L_1)$-smoothness due to the particular structure of the anisotropic descent inequality (4). For example, consider the special case where $\phi = \frac{1}{2}\|\cdot\|^2$ and the main algorithm becomes standard gradient descent with fixed stepsize. Then, the anisotropic descent inequality is just the Euclidean one after completing the squares. By using the Pythagorean theorem, one can handle the terms on the r.h.s. of the inequality and then follow a similar technique to [34]. Nevertheless, such a property is not present for the general reference functions that we consider in this paper and thus a different approach is necessary.

Another difficulty arising due to the generality of the setting we consider is that there do not exist global upper bounds for $\|\nabla f(x) - \nabla f(\bar{x})\|$, in contrast to Euclidean [34] or $(L_0, L_1)$-smoothness [47]. Finally, due to the general nature of the preconditioner $\nabla\phi^*$ which can span the whole space, the analysis becomes more complicated, in contrast to clipping or normalized gradient methods. To better motivate this, note that for normalized gradient type methods such as [30, Algorithm 1], the primal update takes the form $x^{k+1} = x^k - \gamma\frac{m^k}{\|m^k\|}$, and as such $\|x^{k+1} - x^k\| = \gamma$, thus bounding the distance between consecutive iterates and simplifying the analysis.

We are nonetheless able to prove the convergence of the method with a novel proof technique that is solely based on the convexity of the reference function $\phi$, thus also unifying and greatly simplifying the analysis. As a caveat, we are not able to show the result for any $\beta \in [0, 1)$ as is the case for Lipschitz smooth functions [34]. Whether or not this constraint can be lifted for the general setting we consider, is an interesting open question.

## 2.1 Convergence under generalized PL condition

In a majorization-minimization procedure, upper bounds of the cost function are usually utilized to study the general convergence properties of the method, such as asymptotic convergence and a sublinear rate for some optimality measure, while lower bounds are used to prove faster rates of convergence, e.g., linear convergence rates for the suboptimality gap.

One of the major advantages of the dual space preconditioning / anisotropic gradient descent method compared to other frameworks of generalized smoothness is that the aforementioned lower bounds are well-studied. In our setting of smooth nonconvex optimization they take the form of the anisotropic gradient dominance condition from [24, Definition 5.6]:

**Definition 2.3.** *We say that $f$ satisfies the anisotropic gradient dominance condition relative to $\phi$ with constant $\mu > 0$ if for all $x \in \mathbb{R}^n$*

$$\phi(\nabla\phi^*(\nabla f(x))) \geq \mu(f(x) - f_\star). \tag{9}$$

The fact that this condition is a generalization of the classical PL inequality becomes evident when choosing $\phi = \frac{1}{2}\|\cdot\|^2$, where (9) becomes $\frac{1}{2}\|\nabla f(x)\|^2 \geq \mu(f(x) - f_\star)$. Definition 2.3 holds for example when $f$ is anisotropically strongly convex [24, Proposition 5.11], in parallel to the standard Euclidean setting where strong convexity implies the PL inequality. Therefore, in light of [24, Proposition 5.11] it holds when $f^*$ is smooth relative to $\phi^*$ with constant $\mu^{-1}$, i.e., when $\mu^{-1}\phi^* - f^*$ is convex. Examples of functions $f$ that satisfy Definition 2.3 relative to some $\phi$ can be found in [24] and in [32, Section 4].

For the base method, linear convergence under this generalized PL condition was shown in [24, Theorem 5.7] and this result was further refined in [35, Lemma 6.7]. In this paper we manage to extend the aforementioned results, showing that Algorithm 1 also converges linearly when $\phi$ is a 2-subhomogeneous function [36, Theorem 3.7], i.e., $\phi(\theta x) \leq \theta^2\phi(x)$ for all $\theta \in [0,1]$ and $x \in \text{dom}\,\phi$. We provide examples of important 2-subhomogeneous reference functions as well as further discussion on the generalized PL condition in Appendix E.

**Theorem 2.4.** *Let Assumption 1.3 hold and $f$ satisfy the anisotropic gradient dominance condition relative to $\phi$ with constant $\mu > 0$, where $\phi$ is 2-subhomogeneous. Let, moreover, $\{x^k\}_{k\in\mathbb{N}_0}$ be the sequence of iterates generated by Algorithm 1 with $\beta \in (0, 0.5)$ and $\gamma \leq \frac{1}{L}$. Then, we have the following rate:*

$$f(x^k) - f_\star \leq \alpha^k(f(x^0) - f_\star), \tag{10}$$

*where $\alpha = \max\{1 - \gamma\mu(\beta - 2\beta^2), \beta + 2\beta^2\}$.*

In contrast to the proof of Theorem 2.2, in the proof of Theorem 2.4 we utilize a Lyapunov function that is monotonically decreasing along the sequence of iterates, given by $V_k = \gamma\phi(m^{k-1}) + f(x^k) - f_\star$. We further utilize the convexity and 2-subhomogeneity of $\phi$ along with the anisotropic gradient dominance condition to show that $V_k$ converges linearly and thus obtain the claimed result.

## 2.2 Convergence under preconditioned Lipschitz continuity

As already mentioned, one of the main difficulties in proving the convergence of Algorithm 1 comes from the fact that anisotropic smoothness is not equivalent to an upper bound on $\|\nabla f(x) - \nabla f(\bar{x})\|$ for $x, \bar{x} \in \mathbb{R}^n$. Intuitively, this is because the second-order condition for anisotropic smoothness (5) does not involve the spectral norm of the Hessian of $f$, as is for example the case for standard Lipschitz smoothness where $\|\nabla^2 f(x)\| \leq L$ or $(L_0, L_1)$-smoothness where $\|\nabla^2 f(x)\| \leq L_0 + L_1\|\nabla f(x)\|$. Nevertheless, it involves the Jacobian matrix of the preconditioned gradient operator $\nabla\phi^* \circ \nabla f$ and under some generally mild conditions, we can then obtain a bound of the form $\|\nabla(\nabla\phi^* \circ \nabla f)(x)\| \leq L$, which then implies global $L$-Lipschitz continuity of $\nabla\phi^* \circ \nabla f$. We call this condition preconditioned Lipschitz continuity and formally define it in the following assumption.

**Assumption 2.5.** *For any $x, \bar{x} \in \mathbb{R}^n$ the following inequality holds with $L$ as in Assumption 1.3:*

$$\|\nabla\phi^*(\nabla f(x)) - \nabla\phi^*(\nabla f(\bar{x}))\| \leq L\|x - \bar{x}\|. \tag{11}$$

Assumption 2.5 states that the preconditioned gradient is a Lipschitzian operator, hence the name preconditioned Lipschitz continuity. We remark that although more restrictive than anisotropic smoothness itself, this assumption is in fact mild since it holds for example for Lipschitz smooth

functions. Moreover, as we show in the following proposition, it also holds for $f \in \mathcal{C}^2(\mathbb{R}^n)$ that are $(L_0, L_1)$-smooth for a suitable choice of the reference function $\phi$.

**Proposition 2.6.** *Let $f \in \mathcal{C}^2(\mathbb{R}^n)$ be $(L_0, L_1)$-smooth. Then, $f$ satisfies Assumption 2.5 for the reference function $\phi(x) = \frac{L_0}{L_1}(-\|x\| - \ln(1 - \|x\|))$. More precisely, the following inequality holds for all $x, \bar{x} \in \mathbb{R}^n$:*

$$\left\| \frac{\nabla f(x)}{L_0 + L_1 \|\nabla f(x)\|} - \frac{\nabla f(\bar{x})}{L_0 + L_1 \|\nabla f(\bar{x})\|} \right\| \leq \|x - \bar{x}\|. \tag{12}$$

To the best of our knowledge Proposition 2.6 also provides a new characterization for $(L_0, L_1)$-smooth functions. Therefore, Assumption 2.5 is a natural assumption that is at least as general as $(L_0, L_1)$-smoothness. In fact, using arguments similar to those in Remark 1.4, we can show that it is *less restrictive* than $(L_0, L_1)$-smoothness. We now turn to our main result regarding the convergence of Algorithm 1, where we allow $\beta \in (0, 1)$.

**Theorem 2.7.** *Let Assumptions 1.3 and 2.5 hold for $\phi = h \circ \| \cdot \|$. Let, moreover, $\{x^k\}_{k \in \mathbb{N}_0}$ be the sequence of iterates generated by Algorithm 1 with $\beta \in (0, 1)$ and $\gamma = \frac{(1-\beta)^2}{L}$. Then, we have the following rate:*

$$\min_{1 \leq k \leq K} \phi(\nabla \phi^*(\nabla f(x^k))) \leq \frac{1}{K} \left( \frac{f(x^0) - f_\star}{\beta \gamma} + \frac{1}{1 - \beta} \phi(\nabla \phi^*(\nabla f(x^0))) \right). \tag{13}$$

Note that the proofs of Theorems 2.2, 2.4 and 2.7 are based mainly on the convexity of $\phi$ and further utilize that $\phi$ is even. We can obtain even better convergence guarantees by exploiting the strong convexity of $\phi$ as well as the 2-subhomogeneity of important reference functions, but we decide to keep the analysis general enough and easier to follow.

## 3 The stochastic nonlinearly preconditioned gradient method

In this section we study a stochastic version of the base iterate (1). More precisely, we assume that we have access to a stochastic first-order oracle which returns a stochastic gradient $g(x)$, similarly to [22, p. 114 Assumption 2]. The algorithm then takes the following form:

$$x^{k+1} = x^k - \gamma \nabla \phi^*(g(x^k)). \tag{14}$$

Throughout this section we assume that $\operatorname{dom} \phi = \mathbb{R}^n$. The proofs are deferred to Appendix D.

Our first result describes the behavior of the method under a noise condition that, as we demonstrate later on, is less restrictive than bounded variance of the stochastic gradients for many interesting reference functions $\phi$. Note that for now we do not assume that $g$ is an unbiased estimator of $\nabla f$.

**Theorem 3.1.** *Let Assumption 1.3 hold and $\mathbb{E}[\phi(\nabla \phi^*(\nabla f(x)) - \nabla \phi^*(g(x)))] \leq \sigma^2$. Let moreover $\{x^k\}_{k \in \mathbb{N}}$ be the sequence of iterates generated by (14) with stepsize $\gamma \leq \frac{1}{L}$. Then, the following holds:*

$$\mathbb{E}\left[ \frac{1}{K} \sum_{k=0}^{K-1} \phi(\nabla \phi^*(\nabla f(x^k))) \right] \leq \frac{(f(x^0) - f_\star)}{\gamma K} + \sigma^2. \tag{15}$$

The assumption on the stochastic gradients in Theorem 3.1 might seem unintuitive upon first inspection. Nevertheless, it follows naturally from the anisotropic smoothness condition, similarly to how the bounded variance assumption is tailored to Lipschitz smoothness. Moreover, as we show in the next proposition, it is at least as general as the well-established bounded variance assumption for interesting reference functions $\phi$.

**Proposition 3.2.** *Let $\phi$ be either $\sum_{i=1}^n \cosh(x_i) - 1$ or $\cosh(\|x\|) - 1$. Then, the following inequality holds for all $y, \bar{y} \in \mathbb{R}^n$:*

$$\phi(\nabla \phi^*(y) - \nabla \phi^*(\bar{y})) \leq \tfrac{1}{2} \|y - \bar{y}\|^2. \tag{16}$$

*Therefore, if $\mathbb{E}[\|\nabla f(x) - g(x)\|^2] \leq \sigma^2$ we have that*

$$\mathbb{E}[\phi(\nabla \phi^*(\nabla f(x)) - \nabla \phi^*(g(x)))] \leq \frac{\sigma^2}{2}. \tag{17}$$

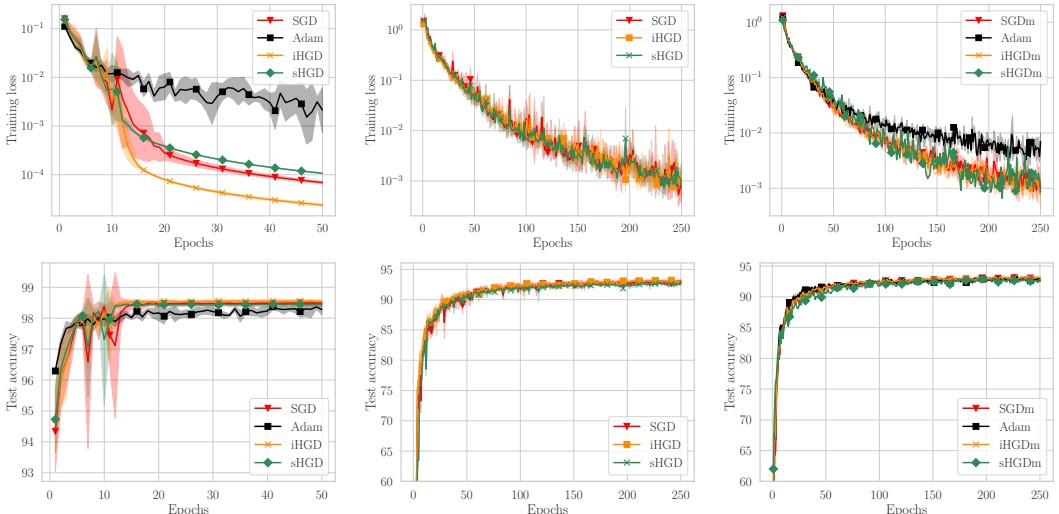

Figure 1: Results for training an MLP on MNIST and ResNet-18 on Cifar10. Top row is the training loss and bottom row the test accuracy. (left) MNIST MLP (middle) Cifar10 ResNet18 without momentum (right) Cifar10 ResNet18 with momentum.

In the proof of Proposition 3.2 we show that there exist even tighter bounds between the two quantities. We next provide a simple example demonstrating that in fact the noise assumption of Theorem 3.1 is less restrictive than bounded variance.

**Example 3.3.** Consider the function $f(x) = \frac{1}{2}[(x-1)^2 + 2(x+2)^2]$ with $x \in \mathbb{R}$. In this case we have $f'_1(x) = 2(x-1)$, $f'_2(x) = 4(x+2)$ and it is straightforward that as $|x| \to +\infty$ also $\mathbb{E}[|f'_i(x) - f'(x)|^2]$ becomes unbounded above. Nevertheless, choosing $\phi = \cosh -1$ we have that $\phi(\phi^{*'}(f'(x)) - \phi^{*'}(f'_i(x)))$ is upper bounded for every sample $i$. It is also clear that the example considered here does not satisfy the $p$-th bounded moment assumption (also known as heavy-tailed noise in the related literature) [15, Assumption 3]. Nevertheless, we are not aware if it is actually a less restrictive assumption, since there does not seem to exist a simple inequality that connects the two quantities.

Although the noise assumption in Theorem 3.1 follows naturally from the analysis of the algorithm, it is not clear whether we can prove the convergence of the method under it. Moreover, due to the nonlinear preconditioning of the proposed method in (14), the difference $x^{k+1} - x^k$ is not a true estimator of $\nabla \phi^* \circ \nabla f$ even when $\mathbb{E}[g(x)] = \nabla f(x)$, i.e. $\mathbb{E}[\nabla \phi^*(g(x))] \neq \nabla \phi^*(\nabla f(x))$, similarly to the gradient clipping method [6, 20]. One way to reduce the potential bias introduced by the preconditioning when $g$ is a true estimator of $\nabla f$, is to increase the batch size. Utilizing thus Proposition 3.2, we obtain the following result.

**Theorem 3.4.** *Let Assumption 1.3 hold for some $\phi$ such that* (16) *holds and assume moreover that* $\mathbb{E}[\|\nabla f(x) - g(x)\|^2] \leq \frac{\sigma^2}{K}$ *for all $x \in \mathbb{R}^n$. If we run* (14) *for $K > 0$ iterations with $\gamma \leq \frac{1}{L}$, we have the following rate.*

$$\mathbb{E}\left[\frac{1}{K}\sum_{k=0}^{K-1} \phi(\nabla\phi^*(\nabla f(x^k)))\right] \leq \frac{1}{K}\left[\frac{(f(x^0) - f_\star)}{\gamma} + \frac{\sigma^2}{2}\right]. \tag{18}$$

The noise assumption in Theorem 3.4 corresponds to the case where the stochastic oracle is unbiased, has bounded variance and we run a minibatch version of (14) with batch size $K$. Therefore, Theorem 3.4 describes convergence guarantees for the method under standard assumptions on the stochastic gradients and a generalized smoothness assumption that goes beyond Lipschitz smoothness for the cost function itself. Moreover, it covers both isotropic reference functions $\phi(x) = h(\|x\|)$ and separable ones $\phi = \sum_{i=1}^n h(x_i)$. We emphasize the importance of this fact because for separable reference functions (14) takes the form of SGD with a coordinate-wise stepsize that depends on the iterates, effectively complicating the analysis but also leading to more interesting algorithms.

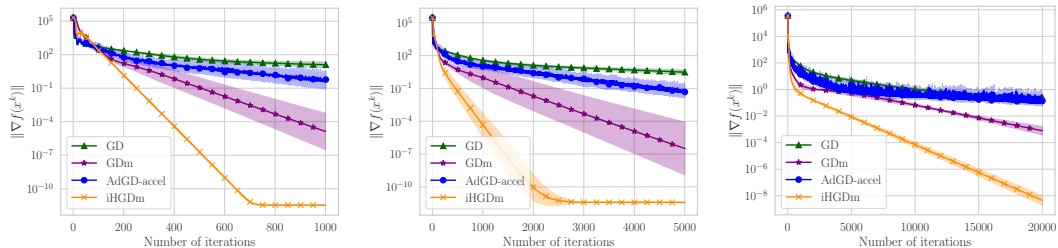

Figure 2: Results for the matrix factorization problem. The figure on the left corresponds to $r = 10$, the one in the middle to $r = 20$ and the one on the right to $r = 30$. It can be seen that our method, iHGDm, significantly outperforms the rest of the methods.

The following result describes the linear convergence of the method up to a constant under the generalized PL condition described in Definition 2.3.

**Theorem 3.5.** *Let Assumption 1.3 hold, $\mathbb{E}[\phi(\nabla\phi^*(\nabla f(x)) - \nabla\phi^*(g(x)))] \leq \sigma^2$ and $f$ satisfy the anisotropic gradient dominance condition relative to $\phi$ with constant $\mu$. Let moreover $\{x^k\}_{k \in \mathbb{N}_0}$ be the sequence of iterates generated by (14) with stepsize $\gamma \leq \frac{1}{L}$. Then, the following holds:*

$$\mathbb{E}[f(x^k) - f_\star] \leq (1 - \gamma\mu)^k(f(x^0) - f_\star) + \frac{\sigma^2}{\mu}. \tag{19}$$

We remark that we have not provided a stochastic extension of the proposed method with momentum Algorithm 1. We believe that this would require specializing to reference functions with more favorable properties and extending results from our stochastic analysis. We leave a detailed investigation of this to future research.

## 4 Experiments

In our experiments, we compare the proposed algorithms with baseline and state-of-the-art methods: stochastic gradient descent (SGD), SGD with momentum (SGDm), Adam [19], gradient descent (GD), gradient descent with momentum (GDm) and the heuristic accelerated variant of the adaptive gradient method from [33] (AdGD-accel). Our methods are versions of Algorithm 1 and (14) that are generated by $\cosh -1$. The code for reproducing the experiments is publicly available[2].

### 4.1 Neural networks

We first consider some neural network experiments. Here, we implemented (14) and the stochastic version of Algorithm 1, while we used the standard PyTorch [38] implementations for SGD and Adam. Moreover, for Adam we used the standard momentum parameters. For all the methods we used fixed stepsizes.

**Image classification on the MNIST dataset [9].** For our first experiment we compare our base method (14) with SGD and Adam on the standard classification problem on the MNIST dataset with the cross-entropy loss. We use a two-layer fully connected neural network with layer dimensions $[28 \times 28, 512, 256]$ and the ReLU activation function and a batch size of 256. We tuned the stepsizes by performing an adaptive gridsearch on one seed and then decreasing the stepsize if the results varied greatly across different seeds. The confidence intervals are obtained from five random seeds. The results are presented in the left column of Figure 1. It can be seen that iHGD reaches small training losses fast and performs better than the rest of the methods.

**Image classification on the Cifar10 dataset [21].** In this experiment we use the standard ResNet-18 [14] architecture and train it to classify images with the cross-entropy loss. We use batch size 128 for all methods.

We first compare the methods without momentum with SGD over five random seeds. The stepsizes are tuned using the same procedure as in the MNIST experiment. The results are presented in

---

[2]https://github.com/JanQ/nonlin-prec-mom-stoch

the middle column of Figure 1. We then compare the proposed methods with momentum with SGDm and Adam. We choose $\beta = 0.9$ for all methods which is also the standard value for training residual networks with SGDm. We set the learning rate by performing a parameter sweep over $\{5, 1, 0.5, 0.1, 0.05, 0.01, 0.005\}$. The results of this experiment are presented in the right column of Figure 1. The confidence intervals for this experiment are obtained from three random seeds. It can be seen that our proposed methods perform similarly to SGD and SGDm which are known to be state-of-the-art for this problem.

### 4.2 Matrix factorization

We consider the matrix factorization problem as presented in [33]: Given a matrix $A \in \mathbb{R}^{m \times n}$ and $r < \min\{m, n\}$ we solve the following minimization problem.

$$\min_{[U,V]=X} f(X) = f(U, V) = \tfrac{1}{2}\|UV^\top - A\|_F^2, \tag{20}$$

where $U \in \mathbb{R}^{m \times r}$ and $V \in \mathbb{R}^{n \times r}$ and $\|\cdot\|_F$ denotes the Frobenius norm. It is important to note that this problem is not convex and also not (globally) Lipschitz smooth, since it is a multivariate polynomial of degree 4. We use the Movielens 100K dataset [13] and values of $r$ in $\{10, 20, 30\}$. For $r = 10$ and $r = 20$ we average the results over 10 random initializations, while for $r = 30$ over 3.

In this experiment we compare iHGDm with GD, GDm and AdGD-accel, which although is designed to tackle convex problems, performs very well on settings beyond convexity as remarked in [33]. For GD we use the same tuning as in [33], which guarantees convergence although the problem is not globally smooth as it involves quartic polynomials. For our method we use $\beta = 0.9$ and stepsize $\gamma = 2$, but consider the method generated by $\phi(x) = \lambda(\cosh(\|x\|) - 1)$ with $\lambda = 100$, thus allowing for two stepsizes, similarly to [36]. It seems that the relation between these two stepsizes plays an important role for the fast convergence of the method. For GDm we use momentum parameter $\beta = 0.9$ and $\gamma = 1/300$. This stepsize can be considered optimal since by doubling the stepsize the method did not converge. The results are presented in Figure 2. It can be seen that iHGDm significantly outperforms the other methods, while it is more stable across the random initializations.

We provide more details about the implementation along with further experimental results in Appendix F.

## 5 Conclusion and future work

In this paper we have analyzed extensions of the dual space preconditioning / anisotropic gradient descent method. We introduced a preconditioned heavy ball-type algorithm and studied its convergence guarantees in the general nonconvex setting and under a generalized PL condition. We moreover presented a stochastic extension of the base method that we analyzed under both new and standard noise assumptions in the nonconvex setting. Finally, we tested the proposed methods and showed their good performance on various tasks from machine learning and optimization.

Interesting future work includes a unified analysis of the momentum algorithm Algorithm 1 under relaxed assumptions on the reference function $\phi$ as well as for any $\beta \in [0, 1)$. Moreover, extending the full proximal gradient-type algorithm from [24] to incorporate momentum is an important task with many potential applications, especially for constrained problems. For the stochastic algorithm, we believe that the analysis should focus on removing the dependence on the batch size and including momentum in the base method. We believe that such an endeavor would require building upon tools from martingale theory, such as obtaining generalizations of the standard concentration inequalities.

## Acknowledgements

This work was supported by the Research Foundation Flanders (FWO) PhD grant 11A8T26N and research projects G081222N, G033822N, G0A0920N; Research Council KUL grant C14/24/103.

The authors thank Emanuel Laude for the inspiration and helpful conversations.

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

# A  Φ-convexity and anisotropic smoothness

In this section we describe the concept of $\Phi$-convexity and discuss its connection to anisotropic smoothness [24, Appendix A].

**Φ-convexity basics.** For the purpose of this paper, $\Phi$-convexity can be considered as an extension of the envelope representation of convex functions [42, Theorem 8.13] which states that a proper, lsc and convex function is the pointwise supremum of its affine supports. By replacing the affine supports with nonlinear ones, we obtain the definition of $\Phi$-convexity for functions:

**Definition A.1** ($\Phi$-convex functions). *Let $X$ and $Y$ be nonempty sets and $\Phi : X \times Y \to \overline{\mathbb{R}}$ a coupling. Let $f : X \to \overline{\mathbb{R}}$. We say that $f$ is $\Phi$-convex on $X$ if there is an index set $\mathcal{I}$ and parameters $(y_i, \beta_i) \in Y \times \overline{\mathbb{R}}$ for $i \in \mathcal{I}$ such that*

$$f(x) = \sup_{i \in \mathcal{I}} \Phi(x, y_i) - \beta_i \quad \forall x \in X. \tag{21}$$

For a definition of $\Phi$-convex sets, the interested reader can check [37, Section 1.4]. From (21) it is clear that if $\Phi = \langle \cdot, \cdot \rangle$, then $f(x) = \sup_{i \in \mathcal{I}} \langle x, y_i \rangle - \beta_i$ and we recover the class of proper, lsc and convex functions from [42, Theorem 8.13]. An important notion in the $\Phi$-convexity framework is that of $\Phi$-conjugacy, which generalizes standard convex conjugacy:

**Definition A.2** ($\Phi$-conjugate functions). *Let $X$ and $Y$ be nonempty sets and $\Phi : X \times Y \to \overline{\mathbb{R}}$ a coupling. Let $f : X \to \overline{\mathbb{R}}$. Then we define*

$$f^{\Phi}(y) = \sup_{x \in X} \Phi(x, y) - f(x), \tag{22}$$

*as the $\Phi$-conjugate of $f$ on $Y$ and*

$$f^{\Phi\Phi}(x) = \sup_{y \in Y} \Phi(x, y) - f^{\Phi}(y), \tag{23}$$

*as the $\Phi$-biconjugate back on $X$.*

Once again it is clear that by choosing $\Phi = \langle \cdot, \cdot \rangle$ we retrieve the definition of convex conjugates [42, Equation 11]. Note that the $\Phi$-biconjugate is defined back on $X$, similarly to how the convex biconjugate is defined on $E$ instead of $E^{**}$ [5, p. 13]. Choosing $\Phi(x, y) = -\frac{1}{2\gamma}\|x - y\|^2$ we get

$$f^{\Phi}(y) = \sup_{x \in \mathbb{R}^n} -\frac{1}{2\gamma}\|x - y\|^2 - f(x) = -\inf_{x \in \mathbb{R}^n} f(x) + \frac{1}{2\gamma}\|x - y\|^2,$$

i.e., the $\Phi$-conjugate is the negative Moreau envelope of $f$, a very important quantity in optimization theory [42, Definition 1.22]. Finally, we present the definition of the $\Phi$-subgradient which generalizes the notion of the standard convex subgradient:

**Definition A.3** ($\Phi$-subgradients). *Let $X$ and $Y$ be nonempty sets and $\Phi : X \times Y \to \overline{\mathbb{R}}$ a coupling. Let $f : X \to \overline{\mathbb{R}}$. Then we say that $y$ is a $\Phi$-subgradient of $f$ at $\bar{x}$ if*

$$f(x) \geq f(\bar{x}) + \Phi(x, y) - \Phi(\bar{x}, y), \tag{24}$$

*for all $x \in X$. We denote by $\partial_{\Phi} f(\bar{x})$ the set of all $\Phi$-subgradients of $f$ at a point $\bar{x} \in X$, which we call the $\Phi$-subdifferential of $f$. When the $\Phi$-subdifferential is nonempty at some point $\bar{x}$, we say that $f$ is $\Phi$-subdifferentiable at $\bar{x}$ and when it is everywhere nonempty, we say that $f$ is $\Phi$-subdifferentiable.*

From the definition above it is clear that if a function has a $\Phi$-subgradient at a point $\bar{x}$, then it is also $\Phi$-convex at this point. The opposite does not hold in general, as there exist $\Phi$-convex functions that have everywhere empty $\Phi$-subdifferentials [10].

**Anisotropic smoothness.** By choosing $\Phi(x, y) = -\frac{1}{L} \star \phi(x - y)$, the connection between anisotropic smoothness and $\Phi$-convexity can be made evident: the anisotropic descent inequality for $f$ (4) is equivalent to the $\Phi$-subgradient inequality for $-f$, (24). From this equivalence it is also straightforward that at any point $x \in \mathbb{R}^n$, $\partial_{\Phi}(-f)(x) = \{x - \frac{1}{L}\nabla\phi^*(\nabla f(x))\}$ and thus the main iterate (1) boils down to taking a $\Phi$-subgradient of $-f$ at $x^k$.

This connection to $\Phi$-convexity actually leads to an interesting envelope representation of anisotropic smoothness. As discussed above, since $-f$ is $\Phi$-subdifferentiable, it is also $\Phi$-convex, i.e.,

$$-f(x) = \sup_{i \in \mathcal{I}} -\frac{1}{L} \star \phi(x - y_i) - \beta_i,$$

meaning that

$$f(x) = \inf_{i \in \mathcal{I}} \tfrac{1}{L} \star \phi(x - y_i) + \beta_i.$$

Therefore, $f$ is the pointwise infimum over a family of nonlinear functions, in parallel to standard Euclidean smoothness where $f$ is the infimum over convex quadratics, a fact that immediately follows from the Euclidean descent inequality. In that sense, anisotropic smoothness is a natural generalization of Euclidean smoothness, that as discussed in the main text, is also less restrictive than $(L_0, L_1)$-smoothness. Moreover, the $\Phi$-convexity of $-f$ implies that

$$-f(x) = (-f)^{\Phi\Phi}(x) = \sup_{y \in \mathbb{R}^n} -\tfrac{1}{L} \star \phi(x - y) - (-f)^{\Phi}(y) = -\inf_{y \in \mathbb{R}^n} \tfrac{1}{L} \star \phi(x - y) + (-f)^{\Phi}(y),$$

i.e., that $f$ can be written as the infimal-convolution or epi-addition [42, Equation 1(12)] between two functions.

The aforementioned envelope representation leads to important relations that, to the best of our knowledge, are absent in other forms of generalized smoothness. To begin with, when $f$ is convex, in light of [24, Proposition 4.1], anisotropic smoothness relative to $\phi$ with constant $L$ is equivalent to the convexity of $f^* - L^{-1}\phi^*$, with the latter being known in the literature as strong convexity of $f^*$ relative to $\phi^*$ [31]. In this setting therefore, anisotropic smoothness and relative (Bregman) strong convexity form a conjugate duality, in parallel to the standard one between Lipschitz smoothness and strong convexity. Except for further highlighting the fact that anisotropic smoothness is a natural generalization of Lipschitz smoothness, this conjugate duality is very useful in obtaining convergence guarantees in the convex setting [36], while the envelope representation is useful in providing calculus results for anisotropic smoothness [24, Section 4.2].

# B  Helper Lemmas and auxiliary results

**Lemma B.1.** *Let $\{\delta_k\}_{k \in \mathbb{N}_0}$ be a nonnegative sequence of reals that satisfies*

$$\delta_{k+1} \leq (1 - \alpha)\delta_k + \theta, \tag{25}$$

*where $\theta > 0$ and $\alpha \in (0, 2)$. Then,*

$$\delta_k \leq |1 - \alpha|^k \delta_0 + \frac{\theta}{1 - |1 - \alpha|}. \tag{26}$$

*Proof.* By using the fact that $(1 - \alpha)\delta_k \leq |1 - \alpha|\delta_k$ and unrolling (25) we have that

$$\delta_k \leq |1 - \alpha|^k \delta_0 + \theta \sum_{i=0}^{k-1} |1 - \alpha|^i \leq |1 - \alpha|^k \delta_0 + \frac{\theta}{1 - |1 - \alpha|},$$

where the second inequality follows by the fact that $\sum_{i=0}^{k-1} |1 - \alpha|^i \leq \sum_{i=0}^{\infty} |1 - \alpha|^i = \frac{1}{1-|1-\alpha|}$, since $|1 - \alpha| < 1$. $\qquad\square$

In our analysis we will also need the following standard result. We remind that 2-subhomogeneity means that $\phi(\theta x) \leq \theta^2 \phi(x)$ for all $\theta \in [0, 1]$ and $x \in \operatorname{dom} \phi$.

**Proposition B.2.** *Let $\phi : \mathbb{R}^n \to \overline{\mathbb{R}}$ be convex and $\phi(0) = 0$. Then, for constants $\lambda_i \geq 0$ and $\{x_i\}_{i=1}^d \in \operatorname{dom} \phi$, the following inequality holds as long as $\lambda := \sum_{i=1}^d \lambda_i \leq 1$:*

$$\phi\left(\sum_{i=1}^d \lambda_i x_i\right) \leq \sum_{i=1}^d \lambda_i \phi(x_i). \tag{27}$$

*If, moreover, $\phi$ is 2-subhomogeneous, then*

$$\phi\left(\sum_{i=1}^d \lambda_i x_i\right) \leq \lambda \sum_{i=1}^d \lambda_i \phi(x_i). \tag{28}$$

*Proof.* The proof of the first statement follows immediately by the convexity inequality along with the fact that $\phi(0) = 0$:

$$\phi\left(\sum_{i=1}^{d}\lambda_i x_i\right) = \phi\left((1-\lambda)0 + \sum_{i=1}^{d}\lambda_i x_i\right) \leq (1-\lambda)\phi(0) + \sum_{i=1}^{d}\lambda_i\phi(x_i) = \sum_{i=1}^{d}\lambda_i\phi(x_i).$$

Regarding the second statement, we have:

$$\phi\left(\sum_{i=1}^{d}\lambda_i x_i\right) = \phi\left(\lambda\sum_{i=1}^{d}\tfrac{\lambda_i}{\lambda}x_i\right) \leq \lambda^2\phi\left(\sum_{i=1}^{d}\tfrac{\lambda_i}{\lambda}x_i\right) \leq \lambda\sum_{i=1}^{d}\lambda_i\phi(x_i).$$

The first inequality follows by the definition of 2-subhomogeneity, since $\sum_{i=1}^{d}\frac{\lambda_i}{\lambda}x_i \in \operatorname{dom}\phi$ and the second one by the convexity inequality. $\qquad\square$

We will also need an extension of the monotonicity property of anisotropic smoothness under episcaling [24, Proposition 4.8] to our setting where $\phi$ is possibly not of full domain.

**Proposition B.3.** *Let* $f \in \mathcal{C}^1(\mathbb{R}^n)$, $L_2 > L_1 > 0$ *and* $x, \bar{x} \in \mathbb{R}^n$, *and suppose that*

$$f(x) \leq f(\bar{x}) + \tfrac{1}{L_1}\star\phi(x - \bar{x} + L_1^{-1}\nabla\phi^*(\nabla f(\bar{x}))) - \tfrac{1}{L_1}\star\phi(L_1^{-1}\nabla\phi^*(\nabla f(\bar{x}))).$$

*Then, it holds that*

$$f(x) \leq f(\bar{x}) + \tfrac{1}{L_2}\star\phi(x - \bar{x} + L_2^{-1}\nabla\phi^*(\nabla f(\bar{x}))) - \tfrac{1}{L_2}\star\phi(L_2^{-1}\nabla\phi^*(\nabla f(\bar{x}))).$$

*Therefore, if $f$ has the anisotropic descent property relative to $\phi$ with constant $L_1 > 0$, then $f$ has the anisotropic descent property relative to $\phi$ for any $L_2 > L_1$.*

*Proof.* We define the shorthand $\bar{u} = \nabla\phi^*(\nabla f(\bar{x}))$ for ease of presentation. Note that if $\operatorname{dom}\phi = \mathbb{R}^n$, then the statement follows from [24, Proposition 4.8].

Now fix any $x, \bar{x} \in \mathbb{R}^n$. If $L_2(x - \bar{x}) + \bar{u} \notin \operatorname{dom}\phi$, the statement holds vacuously. We first show that if $L_2(x - \bar{x}) + \bar{u} \in \operatorname{dom}\phi$, then $L_1(x - \bar{x}) + \bar{u} \in \operatorname{dom}\phi$ as well:

Consider $t \in [0, 1]$. Then, by convexity, $t(L_2(x - \bar{x}) + \bar{u}) + (1 - t)\bar{u} \in \operatorname{dom}\phi$ where we also used that $\bar{u} \in \operatorname{dom}\phi$ by convex conjugacy. Choose $t = \frac{L_1}{L_2} < 1$ and then we have

$$L_1(x - \bar{x}) + \tfrac{L_1}{L_2}\bar{u} + (1 - \tfrac{L_1}{L_2})\bar{u} \in \operatorname{dom}\phi$$
$$\Longleftrightarrow L_1(x - \bar{x}) + \bar{u} \in \operatorname{dom}\phi.$$

Therefore, if $L_1(x - \bar{x}) + \bar{u} \notin \operatorname{dom}\phi$ the statement holds vacuously as well. Moreover, we have the following:

$$\begin{aligned}
\tfrac{1}{L_1}\phi(L_1(x - \bar{x}) + \bar{u}) &= \tfrac{1}{L_1}\phi(\tfrac{L_1}{L_2}(L_2(x - \bar{x}) + \bar{u}) + (1 - \tfrac{L_1}{L_2})\bar{u}) \\
&\leq \tfrac{1}{L_2}\phi(L_2(x - \bar{x}) + \bar{u}) + \tfrac{1}{L_1}(1 - \tfrac{L_1}{L_2})\phi(\bar{u}) \\
&= \tfrac{1}{L_2}\star\phi(x - \bar{x} + L_2^{-1}\bar{u}) + (\tfrac{1}{L_1} - \tfrac{1}{L_2})\phi(\bar{u}),
\end{aligned}$$

where the inequality follows by convexity and the second equality by the definition of episcaling. The claimed result now follows by substituting the inequality above into the anisotropic descent inequality with constant $L_1$. $\qquad\square$

## C   Proofs of Section 2

We define the shorthand $u^k := \nabla\phi^*(\nabla f(x^k))$ for ease of presentation.

## C.1 Proof of Theorem 2.2

*Proof.* To set up our proof let us first unroll the recursion for $m^k$:

$$m^0 = (1 - \beta)u^0$$
$$m^1 = \beta(1 - \beta)u^0 + (1 - \beta)u^1$$
$$m^2 = \beta^2(1 - \beta)u^0 + \beta(1 - \beta)u^1 + (1 - \beta)u^2$$
$$\vdots$$
$$m^k = (1 - \beta) \sum_{k'=0}^{k} \beta^{k'} u^{k-k'}.$$

Note that since $\gamma \leq \frac{1}{L}$, $\frac{1}{\gamma} \geq L$ and we can use Proposition B.3 with constant $\frac{1}{\gamma}$:

$$f(x^{k+1}) \leq f(x^k) + \gamma\phi(\tfrac{1}{\gamma}(x^{k+1} - x^k) + u^k) - \gamma\phi(u^k)$$
$$= f(x^k) + \gamma\phi(u^k - m^k) - \gamma\phi(u^k),$$

where in the equality we have substituted the update rule for $x^{k+1}$. Substituting the results of the recursion for $m^k$ we thus have:

$$f(x^{k+1}) \leq f(x^k) + \gamma\phi\left(\beta u^k - (1 - \beta) \sum_{k'=1}^{k} \beta^{k'} u^{k-k'}\right) - \gamma\phi(u^k).$$

Note now that the constants in front of $u^k$ and $-u^{k-k'}$ inside the $\phi$ in the first line of the above display are $\beta$ and $(1 - \beta)\beta^k$ for $k \geq 1$. Moreover, we have that

$$\beta + \sum_{i=1}^{k}(1 - \beta)\beta^i = \beta + \beta - \beta^2 + \beta^2 - \beta^3 + \cdots + \beta^k - \beta^{k+1} = 2\beta - \beta^{k+1} < 1,$$

since $\beta < 0.5$. Thus, we can apply (27) in the inequality above and obtain for $k \geq 1$

$$f(x^{k+1}) \leq f(x^k) - (1 - \beta)\gamma\phi(u^k) + \gamma(1 - \beta) \sum_{k'=1}^{k} \beta^{k'} \phi(u^{k-k'})$$

where we have also used the fact that $\phi$ is even. Now let us unroll this inequality in order to make our logic clear, noting that for $k = 0$ we immediately obtain the result from the convexity inequality since the argument of $\phi$ is just $\beta u^0$:

$$f(x^1) \leq f(x^0) - \gamma(1 - \beta)\phi(u^0)$$
$$f(x^2) \leq f(x^1) + \gamma(1 - \beta)\beta\phi(u^0) - \gamma(1 - \beta)\phi(u^1)$$
$$f(x^3) \leq f(x^2) + \gamma(1 - \beta)\beta^2\phi(u^0) + \gamma(1 - \beta)\beta\phi(u^1) - \gamma(1 - \beta)\phi(u^2)$$
$$\vdots$$

up until iterate $K > 0$. Summing up the inequalities for $k = 0, \ldots, K$ we thus have:

$$f(x^{K+1}) \leq f(x^0) - \gamma(1 - \beta)\phi(u^K) - \gamma \sum_{k=0}^{K-1}(1 - \beta)(1 - \bar{\beta})\phi(u^k).$$

where $\bar{\beta} := \sum_{i=1}^{K-k} \beta^i$. Now note that $\bar{\beta} \leq \frac{1}{1-\beta} - 1 = \frac{\beta}{1-\beta}$ and thus $1 - \bar{\beta} \geq \frac{1-2\beta}{1-\beta}$. Using the fact that $\phi \geq 0$, we further obtain:

$$f(x^{K+1}) \leq f(x^0) - \gamma(1 - 2\beta) \sum_{k=0}^{K} \phi(u^k). \tag{29}$$

By rearranging and using $f(x^{K+1}) \geq f_\star$ and $\gamma = \frac{\alpha}{L}$ we have:

$$\min_{0 \leq k \leq K} \phi(u^k) \leq \frac{L(f(x^0) - f_\star)}{\alpha(K + 1)(1 - 2\beta)},$$

which is exactly (8). □

## C.2 Proof of Theorem 2.4

To begin with, we define the Lyapunov function:

$$V_k := \gamma\phi(m^{k-1}) + f(x^k) - f_\star. \tag{30}$$

By convexity of $\phi$, the momentum update of Algorithm 1 and the fact that $u^k \in \mathrm{dom}\,\phi$, $m^k \in \mathrm{dom}\,\phi$ (as a convex combination of elements of $\mathrm{dom}\,\phi$), we have that

$$\gamma\phi(m^k) \leq \gamma\beta\phi(m^{k-1}) + \gamma(1-\beta)\phi(u^k). \tag{31}$$

Now consider the anisotropic descent inequality between points $x^k$ and $x^{k+1}$, but with constant $\frac{1}{\gamma} \geq L$, from Proposition B.3:

$$
\begin{aligned}
f(x^{k+1}) &\leq f(x^k) + \gamma\phi(u^k - m^k) - \gamma\phi(u^k) \\
&= f(x^k) + \gamma\phi(\beta(u^k - m^{k-1})) - \gamma\phi(u^k),
\end{aligned}
$$

where we have moreover used the update of $m^k$. Utilizing (28) with $\lambda = 2\beta < 1$ and the fact that $\phi$ is even we can further bound

$$f(x^{k+1}) \leq f(x^k) + 2\beta^2\gamma\phi(u^k) + 2\beta^2\gamma\phi(m^{k-1}) - \gamma\phi(u^k). \tag{32}$$

Denoting $P_k := f(x^k) - f_\star$ and summing (31) and (32) we have:

$$P_{k+1} + \gamma\phi(m^k) \leq P_k + \gamma(\beta + 2\beta^2)\phi(m^{k-1}) - \gamma(\beta - 2\beta^2)\phi(u^k).$$

Note that $\beta + 2\beta^2 < 1$ and $\beta - 2\beta^2 > 0$ by our choice of $0 < \beta < 0.5$. From the anisotropic gradient dominance condition, we have that $\phi(u^k) \geq \mu P_k$ and thus

$$P_{k+1} + \gamma\phi(m^k) \leq (1 - (\beta - 2\beta^2)\gamma\mu)P_k + (\beta + 2\beta^2)\gamma\phi(m^{k-1}).$$

Since $\phi \geq 0$ and $P_k \geq 0$ we can further bound the r.h.s. by

$$V_{k+1} \leq \max\{1 - (\beta - 2\beta^2)\gamma\mu, \beta + 2\beta^2\}V_k.$$

Iterating now this inequality we obtain the claimed result.

## C.3 Proof of Proposition 2.6

To begin with, we prove the following more general result that we utilize later on:

**Proposition C.1.** *Let* $\|\nabla^2\phi^*(\nabla f(x))\nabla^2 f(x)\| \leq L$ *hold for all* $x \in \mathbb{R}^n$*. Then, for all* $x, \bar{x} \in \mathbb{R}^n$:

$$\|\nabla\phi^*(\nabla f(x)) - \nabla\phi^*(\nabla f(\bar{x}))\| \leq L\|x - \bar{x}\|. \tag{33}$$

*Proof.* By the fundamental theorem of calculus, for any $x, \bar{x} \in \mathbb{R}^n$:

$$\nabla\phi^*(\nabla f(x)) - \nabla\phi^*(\nabla f(\bar{x})) = \int_0^1 \nabla^2\phi^*(\nabla f(\bar{x} + t(x - \bar{x})))\nabla^2 f(\bar{x} + t(x - \bar{x}))(x - \bar{x})dt$$

and as such

$$
\begin{aligned}
\|\nabla\phi^*(\nabla f(x)) - \nabla\phi^*(\nabla f(\bar{x}))\| &= \left\|\int_0^1 \nabla^2\phi^*(\nabla f(\bar{x} + t(x - \bar{x})))\nabla^2 f(\bar{x} + t(x - \bar{x}))(x - \bar{x})dt\right\| \\
&\leq \int_0^1 \|\nabla^2\phi^*(\nabla f(\bar{x} + t(x - \bar{x})))\nabla^2 f(\bar{x} + t(x - \bar{x}))\|dt\|x - \bar{x}\| \\
&\leq L\|x - \bar{x}\|,
\end{aligned}
$$

where the second inequality follows by the assumption of the statement. $\square$

Now, regarding the proof of Proposition 2.6, note that since the spectral norm is submultiplicative,

$$\|\nabla^2\phi^*(\nabla f(x))\nabla^2 f(x)\| \leq \|\nabla^2\phi^*(\nabla f(x))\|\|\nabla^2 f(x)\|.$$

We have that $\phi(x) = \frac{L_0}{L_1}\varphi(x)$ where $\varphi(x) := -\|x\| - \ln(1 - \|x\|)$ and by standard convex conjugacy, $\nabla\phi^*(y) = \nabla\varphi^*(y/\frac{L_0}{L_1})$. Since $\nabla\varphi^*(y) = \frac{y}{1+\|y\|}$, we thus have:

$$\nabla\phi^*(y) = \frac{y}{\frac{L_0}{L_1} + \|y\|} = L_1\frac{y}{L_0 + L_1\|y\|}.$$

By simple calculations we then have

$$\nabla^2\phi^*(y) = \frac{1}{\frac{L_0}{L_1} + \|y\|}I + \left(\frac{\frac{L_0}{L_1}}{(\|y\| + \frac{L_0}{L_1})^2} - \frac{1}{\frac{L_0}{L_1} + \|y\|}\right)\frac{yy^\top}{\|y\|^2}$$

$$= \frac{1}{\frac{L_0}{L_1} + \|y\|}I - \frac{\|y\|}{(\frac{L_0}{L_1} + \|y\|)^2}\frac{yy^\top}{\|y\|^2}$$

$$= \frac{1}{\frac{L_0}{L_1} + \|y\|}\left(I - \frac{\|y\|}{\frac{L_0}{L_1} + \|y\|}\frac{yy^\top}{\|y\|^2}\right)$$

The term multiplying $\frac{yy^\top}{\|y\|^2}$ in the display above is therefore negative and as such $\lambda_{\max}(\nabla^2\phi^*(y)) \leq \frac{L_1}{L_0+L_1\|y\|}$. Clearly, $\nabla^2\phi^*$ is positive semidefinite as the Hessian of a convex function and thus $\|\nabla^2\phi^*(\nabla f(x))\| \leq \frac{L_1}{L_0+L_1\|\nabla f(x)\|}$. By $(L_0, L_1)$-smoothness we moreover have $\|\nabla^2 f(x)\| \leq L_0 + L_1\|\nabla f(x)\|$ and thus
$$\|\nabla^2\phi^*(\nabla f(x))\nabla^2 f(x)\| \leq L_1.$$
Using now Proposition C.1 completes our proof.

### C.4  Proof of Theorem 2.7

We denote $\alpha := (1 - \beta)^2$ and start from the anisotropic descent inequality between points $x^k$ and $x^{k+1}$ with constant $\frac{1}{\gamma} > L$, Proposition B.3:

$$f(x^{k+1}) \leq f(x^k) + \gamma\phi(\tfrac{1}{\gamma}(x^{k+1} - x^k) + u^k) - \gamma\phi(u^k)$$
$$= f(x^k) + \gamma\phi(\underbrace{\beta u^k - \beta m^{k-1}}_{\varepsilon^k}) - \gamma\phi(u^k). \tag{34}$$

Now, we work out the recursion for $\varepsilon^k$:

$$\varepsilon^0 = \beta u^0$$
$$\varepsilon^1 = \beta u^1 - \beta(1 - \beta)u^0 = \beta(u^1 - u^0) + \beta^2 u^0$$
$$\varepsilon^2 = \beta u^2 - \beta m^1 = \beta(u^2 - u^1) + \beta^2(u^1 - u^0) + \beta^3 u^0$$
$$\vdots$$
$$\varepsilon^k = \beta^{k+1}u^0 + \sum_{k'=0}^{k-1}\beta^{k'+1}(u^{k-k'} - u^{k-k'-1})$$

Therefore, from the triangle inequality we have that

$$\|\varepsilon^k\| \leq \beta^{k+1}\|u^0\| + \sum_{k'=0}^{k-1}\beta^{k'+1}\|u^{k-k'} - u^{k-k'-1}\|$$

$$\leq \beta^{k+1}\|u^0\| + \sum_{k'=0}^{k-1}\beta^{k'+1}L\|x^{k-k'} - x^{k-k'-1}\|$$

$$= \beta^{k+1}\|u^0\| + \sum_{k'=0}^{k-1}\beta^{k'+1}\alpha\|m^{k-k'-1}\|, \tag{35}$$

where we have used Assumption 2.5, the update of the algorithm and the fact that $\gamma = \frac{\alpha}{L}$.

From convexity of $\phi$ we moreover have

$$\gamma\phi(m^k) \leq \beta\gamma\phi(m^{k-1}) + (1-\beta)\gamma\phi(u^k)$$

Summing up (34) and the inequality above we obtain:

$$f(x^{k+1}) + \gamma\phi(m^k) \leq f(x^k) + \beta\gamma\phi(m^{k-1}) + \gamma\phi(\varepsilon_k) - \beta\gamma\phi(u^k). \tag{36}$$

Now, since $\phi = h \circ \|\cdot\|$ we have $\phi(\varepsilon^k) = h(\|\varepsilon^k\|)$ and thus for all $k \geq 0$ we get from (35) and the fact that $h$ is increasing that:

$$\phi(\varepsilon^k) \leq h\left(\beta^{k+1}\|u^0\| + \sum_{k'=0}^{k-1} \beta^{k'+1}\alpha\|m^{k-k'-1}\|\right)$$

$$\leq \beta^{k+1}\phi(u^0) + \alpha\sum_{k'=0}^{k-1} \beta^{k'+1}\phi(m^{k-k'-1}).$$

Regarding the second inequality, note that from [36, Lemma 1.3], $\|u^0\| = |h^{*'}(\|\nabla f(x^0)\|)| \in \operatorname{dom} h$. Moreover, by definition, for any $t \in \mathbb{N}$, $m^t \in \operatorname{dom}\phi$ as a convex combination of elements of $\operatorname{dom}\phi$. Since, $\phi = h \circ \|\cdot\|$ this implies that $\|m^t\| \in \operatorname{dom} h$ as well and thus all the points in the r.h.s. of the first inequality belong in $\operatorname{dom} h$. Furthermore, by our choice of $\alpha$, $\beta^{k+1} + \alpha\sum_{k'=0}^{k-1}\beta^{k'+1} \leq 1$ and thus the second inequality follows from (27).

Plugging this result back into (36) we obtain:

$$f(x^{k+1}) + \gamma\phi(m^k) \leq f(x^k) + \beta\gamma\phi(m^{k-1}) - \beta\gamma\phi(u^k)$$

$$+ \gamma\left[\beta^{k+1}\phi(u^0) + \alpha\sum_{k'=0}^{k-1} \beta^{k'+1}\phi(m^{k-k'-1})\right]$$

We thus have:

$$f(x^1) + \gamma\phi(m^0) \leq f(x^0) = f(x^0) + \beta\gamma\phi(0) = f(x^0) + \beta\gamma\phi(m^{-1})$$

$$f(x^2) + \gamma\phi(m^1) \leq f(x^1) + \beta\gamma\phi(m^0) + \beta^2\gamma\phi(u^0) + \alpha\beta\gamma\phi(m^0) - \beta\gamma\phi(u^1)$$

$$f(x^3) + \gamma\phi(m^2) \leq f(x^2) + \beta\gamma\phi(m^1) + \beta^3\gamma\phi(u^0) + \alpha\beta\gamma\phi(m^1) + \alpha\beta^2\gamma\phi(m^0) - \beta\gamma\phi(u^2)$$

$$\vdots$$

up until $K > 0$. Summing up these inequalities for $k = 0, \ldots, K$ we obtain

$$f(x^{K+1}) + \gamma\sum_{k=0}^{K} \phi(m^k) \leq f(x^0) + \beta\gamma\sum_{k=0}^{K} \phi(m^{k-1}) - \beta\gamma\sum_{k=1}^{K} \phi(u^k)$$

$$+ \gamma\phi(u^0)\sum_{k=1}^{K} \beta^{k+1} + \alpha\gamma\sum_{k=0}^{K}\sum_{k'=0}^{k-1} \beta^{k'+1}\phi(m^{k-k'-1})$$

and by rearranging

$$f(x^{K+1}) + \sum_{k=0}^{K}\left((\gamma - \beta\gamma)\phi(m^k) - \alpha\gamma\sum_{k'=0}^{k-1}\beta^{k-k'}\phi(m^{k'})\right)$$

$$\leq f(x^0) - \beta\gamma\phi(m^K) + \gamma\phi(u^0)\sum_{k=1}^{K}\beta^{K+1} - \beta\gamma\sum_{k=1}^{K}\phi(u^k),$$

further implying

$$f(x^{K+1}) + \gamma\sum_{k=0}^{K}\phi(m^k)\left(1 - \beta - \alpha\sum_{k'=1}^{K-k}\beta^{k'}\right)$$

$$\leq f(x^0) - \beta\gamma\phi(m^K) + \gamma\phi(u^0)\sum_{k=1}^{K}\beta^{K+1} - \beta\gamma\sum_{k=1}^{K}\phi(u^k).$$

Note that $1 - \beta - \alpha \sum_{k'=1}^{K-k} \beta^{k'} \geq 1 - \beta - (1-\beta)^2 \frac{\beta}{1-\beta} \geq 1 - 2\beta + \beta^2 \geq 0$ and thus the terms on the l.h.s. are positive, since $\phi \geq 0$. Moreover, $\sum_{k=1}^{K} \beta^{k+1} \leq \frac{\beta}{1-\beta}$ and thus rearranging and using $f(x^{K+1}) \geq f_\star$ we obtain:

$$\min_{1 \leq k \leq K} \phi(u^k) \leq \frac{1}{K} \left( \frac{f(x^0) - f_\star}{\beta\gamma} - \phi(m^K) + \frac{1}{1-\beta}\phi(u^0) \right),$$

and by dropping the negative term on the r.h.s. is the claimed result.

## D  Proofs of Section 3

### D.1  Proof of Theorem 3.1

*Proof.* We start from the anisotropic descent inequality between points $x^k$ and $x^{k+1}$, and note that, since $\gamma \leq \frac{1}{L}$, the inequality also holds with constant $\frac{1}{\gamma}$ in light of Proposition B.3:

$$f(x^{k+1}) \leq f(x^k) + \gamma \star \phi(x^{k+1} - x^k + \gamma\nabla\phi^*(\nabla f(x^k))) - \gamma\phi(\nabla\phi^*(\nabla f(x^k)))$$
$$= f(x^k) + \gamma\phi(\nabla\phi^*(\nabla f(x^k)) - \nabla\phi^*(g(x^k))) - \gamma\phi(\nabla\phi^*(\nabla f(x^k))), \qquad (37)$$

where the equality follows by substituting the update (14). Taking conditional expectation we obtain:

$$\mathbb{E}[f(x^{k+1}) \mid x^k] \leq f(x^k) + \gamma\sigma^2 - \gamma\phi(\nabla\phi^*(\nabla f(x^k)))$$

Taking total expectation and summing up the inequality above for $k = 0, \ldots, K-1$, we obtain

$$\sum_{i=0}^{K-1} \mathbb{E}[\phi(\nabla\phi^*(\nabla f(x^k)))] \leq \frac{1}{\gamma}(f(x^0) - \mathbb{E}[f(x^K)]) + \sigma^2 K \leq \frac{1}{\gamma}(f(x^0) - f_\star) + \sigma^2 K,$$

which then implies that

$$\mathbb{E}\left[ \frac{1}{K} \sum_{i=0}^{K-1} \phi(\nabla\phi^*(\nabla f(x^k))) \right] \leq \frac{f(x^0) - f_\star}{\gamma K} + \sigma^2.$$

$\square$

### D.2  Proof of Proposition 3.2

Throughout the proof we will use the following facts about hyperbolic functions:

$$\cosh(\operatorname{arsinh}(a)) = \sqrt{1 + a^2} \qquad (38)$$

and

$$\cosh(a - b) = \cosh(a)\cosh(b) - \sinh(a)\sinh(b). \qquad (39)$$

*Proof.* **Case 1:** $\phi(x) = \sum_{i=1}^{n} \cosh(x_i) - 1$. Let $y, \bar{y} \in \mathbb{R}^n$ and note that

$$\phi(\nabla\phi^*(y) - \nabla\phi^*(\bar{y})) = \sum_{i=1}^{n} \cosh(\operatorname{arsinh}(y_i) - \operatorname{arsinh}(\bar{y}_i)) - 1$$

and thus we can work with each individual summand. Therefore, we will prove that for any $a, b \in \mathbb{R}$, $\cosh(\operatorname{arsinh}(a) - \operatorname{arsinh}(b)) - 1 \leq \frac{1}{2}(a-b)^2$. From (38) and (39) we have that

$$\cosh(\operatorname{arsinh}(a) - \operatorname{arsinh}(b)) = \sqrt{1 + a^2}\sqrt{1 + b^2} - ab.$$

Therefore, we need to show that

$$\sqrt{1 + a^2}\sqrt{1 + b^2} - ab - 1 \leq \tfrac{1}{2}(a-b)^2$$
$$\Longleftrightarrow \sqrt{1 + a^2 + b^2 + a^2 b^2} \leq \tfrac{1}{2}a^2 + \tfrac{1}{2}b^2 + 1$$
$$\Longleftrightarrow 1 + a^2 + b^2 + a^2 b^2 \leq \tfrac{1}{4}a^4 + \tfrac{1}{4}b^4 + 1 + \tfrac{1}{2}a^2 b^2 + a^2 + b^2$$
$$\Longleftrightarrow 0 \leq (\tfrac{1}{2}a^2 - \tfrac{1}{2}b^2)^2,$$

which holds trivially. We thus have

$$\phi(\nabla\phi^*(y) - \nabla\phi^*(\bar{y})) \leq \frac{1}{2}\sum_{i=1}^{n}(y_i - \bar{y}_i)^2 = \frac{1}{2}\|y - \bar{y}\|^2.$$

**Case 2:** $\phi(x) = \cosh(\|x\|) - 1$. Note that $\nabla\phi^*(y) = \text{arsinh}(\|y\|)\overline{\text{sign}}(y)$ is a bijection from $\mathbb{R}^n$ to $\mathbb{R}^n$ with inverse $\nabla\phi(x) = \sinh(\|x\|)\overline{\text{sign}}(x)$ and the result holds true if the following equivalent inequality holds true:

$$\phi(x - \bar{x}) \leq \frac{1}{2}\|\nabla\phi(x) - \nabla\phi(\bar{x})\|^2 \qquad \forall x, \bar{x} \in \mathbb{R}^n.$$

In fact, we will show the tighter inequality

$$\phi(x - \bar{x}) + \frac{1}{2}(\phi(x) - \phi(\bar{x}))^2 \leq \frac{1}{2}\|\nabla\phi(x) - \nabla\phi(\bar{x})\|^2. \tag{40}$$

Substituting the definition of $\phi$ and $\nabla\phi$ we equivalently have:

$$\cosh(\|x - \bar{x}\|) - 1 + \frac{1}{2}\cosh(\|x\|)^2 - \cosh(\|x\|)\cosh(\|\bar{x}\|) + \frac{1}{2}\cosh(\|\bar{x}\|)^2$$

$$\leq \frac{1}{2}\sinh(\|x\|)^2 - \sinh(\|x\|)\sinh(\|\bar{x}\|)\frac{\langle x, \bar{x}\rangle}{\|x\|\|\bar{x}\|} + \frac{1}{2}\sinh(\|\bar{x}\|)^2.$$

Using the hyperbolic identity $\cosh(a)^2 - \sinh(a)^2 = 1$ we arrive at

$$\cosh(\|x - \bar{x}\|) \leq \cosh(\|x\|)\cosh(\|\bar{x}\|) - \sinh(\|x\|)\sinh(\|\bar{x}\|)\frac{\langle x, \bar{x}\rangle}{\|x\|\|\bar{x}\|}.$$

Now we can parametrize the expression above by defining $\theta := \frac{\langle x, \bar{x}\rangle}{\|x\|\|\bar{x}\|}$ and

$$l(\theta) := \|x - \bar{x}\| = \sqrt{\|x\|^2 + \|\bar{x}\|^2 - 2\|x\|\|\bar{x}\|\theta}.$$

To this end, we define the function $g : [-1, 1] \to \mathbb{R}$ by

$$g(\theta) := \cosh(\sqrt{\|x\|^2 + \|\bar{x}\|^2 - 2\|x\|\|\bar{x}\|\theta}) - \cosh(\|x\|)\cosh(\|\bar{x}\|) - \sinh(\|x\|)\sinh(\|\bar{x}\|)\theta$$

for which we have that $g(-1) = g(1) = 0$ by (39). By taking the second derivative we get

$$g''(\theta) = \frac{\|x\|^2\|\bar{x}\|^2\cosh(l(\theta))}{l(\theta)^2} - \frac{\|x\|^2\|\bar{x}\|^2\sinh(l(\theta))}{l(\theta)^3}$$

which is nonnegative by virtue of the inequality $z\cosh(z) - \sinh(z) \geq 0$ for all $z \in \mathbb{R}_+$. Therefore $g(\theta)$ is convex, implying that $g(\theta) \leq 0$ for all $\theta \in [-1, 1]$, which in turn implies the claimed result. $\qquad\square$

### D.3   Proof of Theorem 3.4

*Proof.* Recall from (37) that since $\frac{1}{\gamma} \geq L$, it follows from Proposition B.3 that

$$f(x^{k+1}) \leq f(x^k) + \gamma\phi(\nabla\phi^*(\nabla f(x^k)) - \nabla\phi^*(g(x^k))) - \gamma\phi(\nabla\phi^*(\nabla f(x^k))).$$

From (16) we have that

$$\mathbb{E}[\phi(\nabla\phi^*(\nabla f(x^k)) - \nabla\phi^*(g(x^k)))] \leq \frac{1}{2}\mathbb{E}\left[\|\nabla f(x^k) - g(x^k)\|^2\right] \leq \frac{\sigma^2}{2K}$$

and the rest of the proof follows similarly to the proof of Theorem 3.1. $\qquad\square$

### D.4   Proof of Theorem 3.5

*Proof.* Recall from (37) that since $\frac{1}{\gamma} \geq L$, it follows from Proposition B.3 that

$$f(x^{k+1}) \leq f(x^k) + \gamma\phi(\nabla\phi^*(\nabla f(x^k)) - \nabla\phi^*(g(x^k))) - \gamma\phi(\nabla\phi^*(\nabla f(x^k))).$$

Using Definition 2.3, taking conditional expectation and utilizing the assumption on the noise of the stochastic gradient we obtain

$$\mathbb{E}[f(x^{k+1}) \mid x^k] \leq f(x^k) + \gamma\sigma^2 - \gamma\mu(f(x^k) - f_\star).$$

Rearranging and taking total expectations we thus arrive at

$$\mathbb{E}[f(x^{k+1}) - f_\star] \leq (1 - \gamma\mu)\mathbb{E}[f(x^k) - f_\star] + \gamma\sigma^2.$$

The claimed result now follows from Lemma B.1 for $\delta_k = \mathbb{E}[f(x^k) - f_\star]$ and $\theta = \gamma\sigma^2$. $\qquad\square$

# E The anisotropic gradient dominance condition and 2-subhomogeneity

In this section we provide a discussion on the anisotropic gradient dominance condition Definition 2.3 and provide examples of 2-subhomogeneous reference functions. To begin with, the relation of Definition 2.3 with the classical PL condition varies with the properties of the reference function $\phi$. This is captured in the following proposition.

**Proposition E.1.** *Let $\phi : \mathbb{R}^n \to \overline{\mathbb{R}}$ be as in Assumption 1.2 and let $\mu_\phi$ be the strong convexity parameter of $\phi$.*

1. *Then $\phi(\nabla\phi^*(y)) \leq \frac{1}{2\mu_\phi}\|y\|^2$ for all $y \in \mathbb{R}^n$.*

2. *If, moreover, $\phi$ is $L_\phi$-Lipschitz smooth, then $\phi(\nabla\phi^*(y)) \geq \frac{1}{2L_\phi}\|y\|^2$ for all $y \in \mathbb{R}^n$.*

*Proof.* To begin with, note that since $\phi$ is $\mu_\phi$-strongly convex, $\phi^*$ is $\frac{1}{\mu_\phi}$-Lipschitz smooth. Moreover, since $\min\phi = 0$, from [42, Theorem 11.8], $\phi^*(0) = 0$. By the definition of the convex conjugate we have $\phi^*(y) = \langle\nabla\phi^*(y), y\rangle - \phi(\nabla\phi^*(y))$ and as such

$$\phi(\nabla\phi^*(y)) = \langle\nabla\phi^*(y), y\rangle - \phi^*(y) \leq \frac{1}{2\mu_\phi}\|y\|^2,$$

where the inequality follows from the descent lemma for $\phi^*$ between the points $0$ and $y$, and $\phi^*(0) = 0$.

Now, regarding the second item, note that $\phi^*$ is $\frac{1}{L_\phi}$-strongly convex. From the strong convexity inequality we thus have

$$\phi^*(0) \geq \phi^*(y) + \langle\nabla\phi^*(y), -y\rangle + \frac{1}{2L_\phi}\|y\|^2$$

for all $y \in \mathbb{R}^n$. The result now follows from the same arguments as the proof of the first item. $\square$

Therefore, for functions $\phi$ that grow faster than $\frac{1}{2}\|x\|^2$, the generalized PL inequality of Definition 2.3 is in fact stricter than the standard PL inequality. Nevertheless, when considering a local regime, one can translate bounds from one inequality to the other as presented in the following example.

**Example E.2.** Let $\phi(x) = \cosh(x) - 1$. Then, for all $(y, \beta) \in \mathbb{R}^n \times \mathbb{R}_{++}$ such that $\|y\| \leq \beta$ we have that

$$\phi(\nabla\phi^*(y)) \geq \frac{\sqrt{1+\beta^2} - 1}{\beta^2}\|y\|^2$$

*Proof.* Consider the function $l(t) := \sqrt{1+t^2} - 1 - \frac{\sqrt{1+\beta^2}-1}{\beta^2}t^2$ for $t > 0$. It is straightforward that $l(\beta) = l(0) = 0$ and that $\beta, 0$ are the only solutions of $l(t) = 0$ in $[0, \beta]$. Therefore, if we show that for some $\bar{t} \in (0, \beta)$, $l(\bar{t}) > 0$ we are done, since $l$ is a continuous function. Choose $\bar{t} = \beta/2$ and note that

$$l(\beta/2) = \sqrt{1 + \frac{\beta^2}{4}} - 1 - \frac{\sqrt{1+\beta^2} - 1}{4}.$$

Then, for the function $\tilde{l}(\beta) := \sqrt{1 + \frac{\beta^2}{4}} - 1 - \frac{\sqrt{1+\beta^2}-1}{4}, \beta \geq 0$, we have that $\tilde{l}(0) = 0$ and

$$\tilde{l}'(\beta) = \frac{\beta}{4}\left(\frac{1}{\sqrt{\frac{\beta^2}{4}+1}} - \frac{1}{\sqrt{\beta^2+1}}\right) > 0,$$

implying that $\tilde{l}(\beta) > 0$ and thus that $l(\beta/2) > 0$. This finishes the proof. $\square$

We next move on to providing some examples of 2-subhomogeneous functions. To begin with, in light of [36, Lemma 3.8], $\cosh -1$ is a 2-subhomogeneous function. We moreover have the following new results:

**Proposition E.3.** *The functions $h_1(x) = -|x| - \ln(1 - |x|)$ and $h_2(x) = 1 - \sqrt{1 - x^2}$ are 2-subhomogeneous.*

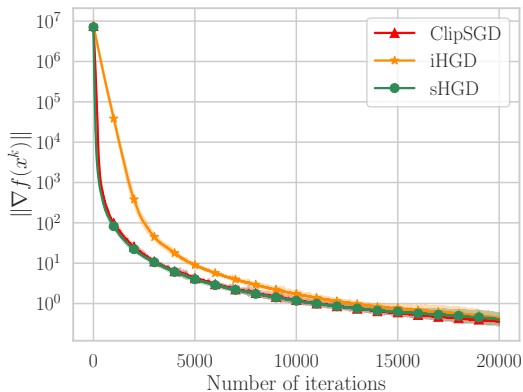

Figure 3: Results for the stochastic implementation of the phase retrieval problem (42).

*Proof.* For $i \in \{1, 2\}$ we want to show the following inequality:

$$h_i(\theta x) \le \theta^2 h_i(x), \tag{41}$$

for all $\theta \in [0, 1]$ and $x \in \operatorname{dom} h_i$. The inequality holds with equality for $\theta = 0$ and $\theta = 1$, so we study $\theta \in (0, 1)$. Now fix $\theta \in (0, 1)$ and consider the function $\tilde{h}_i(x) := h_i(\theta x) - \theta^2 h_i(x)$. We will show that $\tilde{h}_i \le 0$.

For $h_1$ we have $\operatorname{dom} h_1 = (-1, 1)$ and since it is even we consider $x \in [0, 1)$. Clearly, $\tilde{h}_1(0) = 0$ and thus if we show that $\tilde{h}_1'(x) \le 0$ we are done. For $x \in [0, 1)$ and $\theta \in (0, 1)$, we have

$$\tilde{h}_1'(x) = \frac{\theta^2(\theta - 1)x^2}{(x - 1)(\theta x - 1)}.$$

Using $\theta \in (0, 1)$ yields $\tilde{h}_1'(x) \le 0$ for all $x \in [0, 1)$ implying that $\tilde{h}_1$ is a decreasing function and thus $\tilde{h}_1(x) \le \tilde{h}_1(0) = 0$.

For $h_2(x) = 1 - \sqrt{1 - x^2}$, with similar reasoning as above we have

$$\tilde{h}_2'(x) = \frac{\theta^2 x}{\sqrt{1 - \theta^2 x^2}\sqrt{1 - x^2}}\left(\sqrt{1 - x^2} - \sqrt{1 - (\theta x)^2}\right).$$

Clearly, since $\theta \in (0.1)$, $\tilde{h}_2'(x) < 0$ and the proof follows similarly to the proof for $h_1$. □

# F   Further experimental details

## F.1   Experimental details

We provide further details on the experiments presented in Section 4. All neural networks experiments were carried out on an NVIDIA P100 GPU in an internal cluster. The experiment on the MNIST dataset requires less than 2 hours for each algorithm. For the ResNet-18 in the Cifar10 experiments, each run of each algorithm requires approximately 3 hours. The matrix factorization experiments were run on a Intel Core i7-11700 @ 2.50GHz CPU and the total time required was less than 3 hours.

The stepsizes used in the experiments of Subsection 4.1 are presented in Table 2.

Table 2: Stepsizes of the NN experiments in Section 4.

|  | iHGD | sHGD | SGD | Adam |
|---|---|---|---|---|
| MNIST | 1.0 | 0.40 | 0.56 | 0.001 |
| Cifar10 | 0.2 | 0.35 | 0.10 | - |
| Cifar10 momentum | 0.5 | 0.10 | 0.05 | 0.001 |

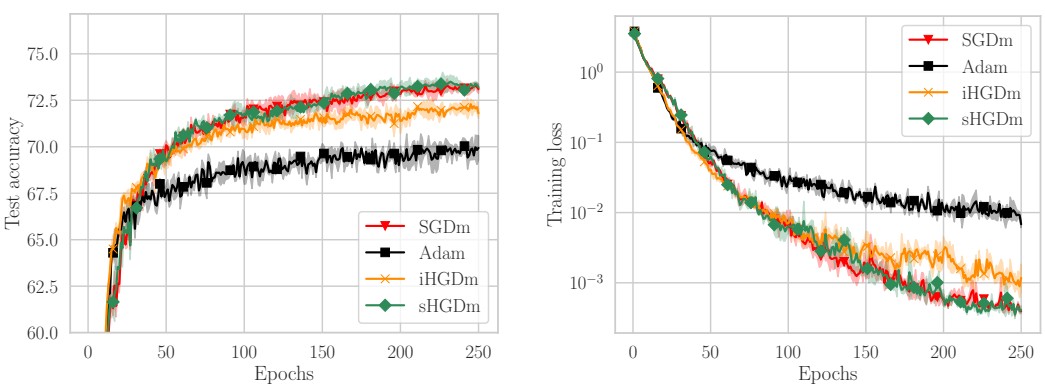

Figure 4: Results for training ResNet-34 on the Cifar100.

## F.2 Additional experiments

### F.2.1 Nonconvex phase retrieval

In this experiment we consider the phase retrieval problem

$$\min f(x) := \frac{1}{2m} \sum_{i=1}^{m} (y_i - (a_i^\top x)^2)^2, \tag{42}$$

where $y_i \in \mathbb{R}$ and $\alpha_i \in \mathbb{R}^n$. We consider an overparametrized problem, with $n = 1000$ and $m = 300$ and $a_i, z \sim \mathcal{N}(0, 0.5)$, $x_0 \sim \mathcal{N}(5, 0.5)$ are generated element-wise with $z$ denoting the ground truth. The measurements are generated as $y_i = (a_i^\top z)^2 + n_i$ with $n_i \sim \mathcal{N}(0, 4^2)$.

We compare the stochastic algorithm (14) generated by $\phi_1(x) = 1000(\cosh(\|x\|) - 1)$ and $\phi_2(x) = 1000 \sum_{i=1}^{n} (\cosh(x_i) - 1)$ with the stochastic clipped gradient method, as presented in [47] with $\nu = 0$. We choose a batch size of $50$ and carefully tune the methods such that they perform well on 10 random problems. For iHGD we choose $\gamma = 1/45$, for sHGD $\gamma = 1/44$ and for the clipped gradient method $\eta = 0.000023$ and $\gamma = 1$ (in the notation of [47]). The results are presented in Figure 3. It can be seen that sHGD performs similarly to the clipped gradient method, while iHGD takes more iterations in the beginning but behaves similarly to the other methods towards the end.

### F.2.2 Image classification on the Cifar100 dataset

In this experiment we use the ResNet-34 architecture and train it to classify images with the cross-entropy loss. We use batch size 128 for all methods. We used stepsizes 1, 0.1, 0.01, 0.001 respectively for iHGDm, sHGDm, SGDm and Adam. The results are presented in Figure 4. The confidence intervals for this experiment are obtained from three random seeds, while for each algorithm one run takes approximately 5 hours.

