# OpenReview forum: "Nonlinearly Preconditioned Gradient Methods: Momentum and Stochastic Analysis"
_NeurIPS.cc/2025/Conference — NeurIPS 2025 poster_

### Official Review · Reviewer_tUyH · 2025-06-21

**Clarity:** 2
**Significance:** 2
**Originality:** 2
**Rating:** 4
**Confidence:** 2

**Summary:**

This paper considers preconditioned gradient methods for nonconvex optimization problems under the generalized smoothness condition. It first introduces preconditioned gradient descent with momentum when the full gradient is available. In addition, the authors propose a stochastic variant of the preconditioned gradient descent method and study its convergence guarantee.

**Questions:**

1. With an appropriate choice of the reference function $\phi$, anisotropic smoothness is equivalent to $(L_0, L_1)$-smoothness. The $(L_0, L_1)$-smoothness framework already encompasses a wide range of machine learning applications. Therefore, a key concern is the significance of further generalizing from $(L_0, L_1)$-smoothness to anisotropic smoothness in the context of machine learning. Specifically, I would like to know which machine learning applications fall within the class of anisotropically smooth functions but not within the class of $(L_0, L_1)$-smooth functions.

**Ethical Concerns:**

["NO or VERY MINOR ethics concerns only"]

**Final Justification:**

The author addressed my concerns. I am satisfied with their response.

**Limitations:**

The theoretical guarantees for the stochastic variants of preconditioned gradient descent appear to be limited, as highlighted in Weaknesses (2) and (3).

**Paper Formatting Concerns:**

No formatting concern.

**Quality:**

2

**Strengths And Weaknesses:**

Strengths:

* Extension of the preconditioned gradient descent to the stochastic setting is an interesting topic to explore.
* The authors provide a rigorous theoretical analysis of the proposed methods.
* This paper presents empirical evidence to demonstrate the effectiveness of the proposed methods.

Weakness:

* The authors do not provide a comparison of the convergence rates of the proposed methods with existing approaches under the generalized smoothness setting. The theoretical advantages of the proposed methods over existing approaches are not clearly articulated.

* In Theorem 3.1, the R.H.S. of (15) has a non-vanishing term of $\sigma^2$, which implies that the proposed stochastic preconditioned gradient descent cannot have exact convergence to the stationary point.

* In Theorem 3.4, the choice of batch size is too large. To obtain $\epsilon$-stationary point, one needs to choose a batch size $K = 1/ \epsilon$, which is impractical for large-scale machine learning applications.

---

> ### Author Rebuttal · Authors · 2025-07-31
>
> We appreciate the reviewer’s time and effort in providing feedback.
>
> In the following we address the **Weaknesses** stated by the reviewer.
>
> > The authors do not provide a comparison of the convergence rates of the proposed methods with existing approaches under the generalized smoothness setting. The theoretical advantages of the proposed methods over existing approaches are not clearly articulated.
>
> We would like to stress that the notion of smoothness that we work with in this paper is different from that of other papers in the generalized smoothness framework. Therefore it is not straightforward to compare the obtained convergence rates except for specific cost functions. Nevertheless, a major advantage of our work is that it provides a unified analysis for a family of algorithms, thus eliminating the need to produce new results for every different setting.
>
> > In Theorem 3.1, the R.H.S. of (15) has a non-vanishing term of $\sigma^2$, which implies that the proposed stochastic preconditioned gradient descent cannot have exact convergence to the stationary point.
>
> Indeed, in Theorem 3.1 we show convergence up to $\sigma^2$ under a new noise assumption that is less restrictive than bounded variance and when the stochastic estimator is potentially biased. We believe that this is an interesting standalone result (such as [3, Section 4.1]), but it also helps provide some insight on our main result, Theorem 3.4 (see also our reply to reviewer 28hb).
>
> > In Theorem 3.4, the choice of batch size is too large. To obtain $\epsilon$-stationary point, one needs to choose a batch size $K=1/\epsilon$, which is impractical for large-scale machine learning applications.
>
> Theorem 3.4 requires large batch sizes to obtain convergence up to $\epsilon$, which to the best of our knowledge is standard for all clipping-type methods under (generalized) smoothness that do not utilize some variance reduction technique or momentum (see [1, Section 3.1], [2, Section 2], [3, Section 4.1]). Using such a technique to remove the need for a large batch size, possibly under a weaker noise assumption, is an important research direction that we believe deserves separate work.
>
> In what follows we answer the **Question** asked by the reviewer.
>
> > 1\. With an appropriate choice of the reference function $\phi$, anisotropic smoothness is equivalent to $(L_0, L_1)$-smoothness. The $(L_0, L_1)$-smoothness framework already encompasses a wide range of machine learning applications. Therefore, a key concern is the significance of further generalizing from $(L_0, L_1)$-smoothness to anisotropic smoothness in the context of machine learning. Specifically, I would like to know which machine learning applications fall within the class of anisotropically smooth functions but not within the class of $(L_0, L_1)$-smooth functions.
>
> We would like to stress that although a $\mathcal{C}^2$ function that is $(L_0, L_1)$-smooth is also anisotropically smooth relative to a suitably chosen reference function, the corresponding descent inequalities are in general not equivalent. In fact for this specific $\phi$, anisotropic smoothness is **less restrictive** than $(L_0, L_1)$-smoothness, as we detail in our answer to reviewer VEKm. Moreover, to the best of our knowledge, although $(L_0, L_1)$-smoothness has been shown **empirically** to better describe some machine learning applications, proving these empirical findings remains an open question.
>
> Regarding the significance of anisotropic smoothness as a notion of generalized smoothness, we would like to stress that as we state in the introduction (lines 42-46) our goal is not only to obtain the most general descent inequality but also to fit the best one to the cost function at hand. From a majorization-minimization perspective we want to find a tight model of the cost function in order to obtain faster algorithms that do not require additional tuning. Therefore, similarly to how $L$-smooth functions also belong to the class of $(L_0, L_1)$-smooth functions (for $L_1 \geq 0$), but the standard approach to minimizing them is standard GD with stepsize $1/L$, there exist functions that are both anisotropically smooth and $(L_0, L_1)$-smooth, but the anisotropic descent inequality describes a tighter model and thus the best approach is the nonlinearly preconditioned gradient method. We next provide some simple examples of 1d functions that are provably anisotropically and $(L_0, L_1)$-smooth, to better motivate this.
> * The function $f(x) = \cosh(x-a)$ is both anisotropically and $(L_0, L_1)$-smooth, but the method described in Equation (1) of the paper for $\phi(x) = \cosh(x)-1$ converges in just one iteration.
> * The function $f(x) = \tfrac{1}{4}x^4 - \tfrac{1}{2}x^2$ is a simple 1d nonconvex function that is anisotropically smooth relative to $\phi(x) = \cosh(x)-1$ with $L = 3.032$ and thus we can use HGD to minimize it. It is also $(L_0, L_1)$-smooth with constants $L_0 = \max |3x^2-1| - L_1|x^3-x|$, where we can freely set $L_1$. We can thus use the algorithm described in [4, Equation (3.2)] to minimize this function. We compare the two methods and present the results in the following table, where the number denotes the required iterations to achieve accuracy $10^{-06}$.
>
> | Initial point | HGD | [4, Equation (3.2)] $L_0=17.02$, $L_1=0.5$ | [4, Equation (3.2)] $L_0=5.08$, $L_1=1$ |
> |---------------|-----|-----------------------------------------|-----------------------------------------|
> | $x_0=10$      | 10  | 25                                      | 55                                      |
> | $x_0=50$      | 21  | 83                                      | 84                                      |
> | $x_0=100$     | 32  | 155                                     | 120                                     |
>
> These simple examples showcase the existence of functions that are better described by the anisotropic descent inequality.
>
> We believe that we have addressed all the reviewer's concerns and weaknesses, and we kindly ask the reviewer to reevaluate their rating.
>
> ---
> **References:**
>
> [1] Hübler, Florian, Ilyas Fatkhullin, and Niao He. "From gradient clipping to normalization for heavy tailed sgd." arXiv preprint arXiv:2410.13849 (2024).
>
> [2] Kornilov, Nikita, et al. "Sign Operator for Coping with Heavy-Tailed Noise in Non-Convex Optimization: High Probability Bounds Under $(L_0, L_1) $-Smoothness." arXiv preprint arXiv:2502.07923 (2025).
>
> [3] Yang, Junchi, et al. "Two sides of one coin: the limits of untuned SGD and the power of adaptive methods." Advances in Neural Information Processing Systems 36 (2023): 74257-74288.

---

> > ### Comment · Reviewer_tUyH · 2025-08-02
> >
> > I have some extra questions:
> >
> > * Regarding the response to my question, what is the reference of [4]?
> >
> > * You mention that *From a majorization-minimization perspective we want to find a tight model of the cost function in order to obtain faster algorithms that do not require additional tuning*. Is there any theoretical justification to show that your algorithm is faster than other algorithms?
> >
> > * You mention that *Therefore it is not straightforward to compare the obtained convergence rates except for specific cost functions.* Could you provide some examples of the specific functions, and compare the obtained convergence rates with existing methods?

---

> > > ### Author Response · Authors · 2025-08-06
> > >
> > > Dear reviewer tUyH
> > >
> > > > Regarding the response to my question, what is the reference of [4]?
> > >
> > > See below for the reference of [4]. We apologize for the omission.
> > >
> > > > You mention that *From a majorization-minimization perspective we want to find a tight model of the cost function in order to obtain faster algorithms that do not require additional tuning.* Is there any theoretical justification to show that your algorithm is faster than other algorithms?
> > >
> > > We would like to stress that our analysis incorporates a generalized PL condition under which we obtain global linear rates with constants that do not depend on the initialization. This implies that our algorithm is in fact provably faster than existing methods for various functions, where the methods for $(L_0, L_1)$ have sublinear rates or at most linear rates with constants depending on the initialization (see for example [5]). Moreover for various examples, our obtained rates are better than those in the related literature, see also our reply to the next question.
> > >
> > > > You mention that *Therefore it is not straightforward to compare the obtained convergence rates except for specific cost functions*. Could you provide some examples of the specific functions, and compare the obtained convergence rates with existing methods?
> > >
> > > In the following we answer the remaining questions. To begin with, since anisotropic smoothness and $(L_0, L_1)$-smoothness form **different** classes of functions, the cost functions have to be specified in order to compare the obtained convergence guarantees. More concretely, consider the reference function $\phi(x) = \cosh(\|x\|)-1$. Then, the convergence guarantees for the stationarity measure $\phi(\nabla \phi^*(\nabla f(x^k)))$ can be translated to the standard one, $\\|\nabla f(x^k)\\|$ and we obtain standard sublinear rates $O(1/\sqrt{k})$ albeit with different constants than those of the $(L_0, L_1)$-literature. We next provide some standard examples from the $(L_0, L_1)$-literature, where our methods are **provably faster**.
> > > 1. $f_1(x) = \cosh(x)$ (which can be found in [6, Theorem 1]). This function is 1-anisotropically smooth w.r.t. $\phi(x) = \cosh(x)-1$ and satisfies the generalized PL inequality with constant 1. This implies the convergence of the base method in Equation (1) in one iteration and of our method Algorithm 1 with a linear rate from Theorem 2.3. The same can be said for $f_2(x) = \exp(|x|)-|x|-1$ w.r.t. $\phi(x) = f_2(x)$. Moreover, $f_2$ is also anisotropically smooth w.r.t. $f_1$ with constant $\sqrt{2}$ and satisfies the generalized PL inequality, thus implying a global linear convergence rate. Since $f_1$ and $f_2$ are strongly convex functions, the analysis of [5] also applies but leads to linear convergence constants that depend on the initialization.
> > > 2. Consider any 1d polynomial function that is $(L_0, L_1)$-smooth, i.e. $|f’’(x)| \leq L_0 + L_1|f’(x)|$ (found e.g. in [7, Lemma 2]). Then it is anisotropically smooth w.r.t. $\phi(x) = L_0/L_1 (\exp(|x|)-|x|-1)$, with constant $L = L_1$. Using simple algebra, we find that our Theorem 2.1 describes the following optimal (choosing $\alpha = 1$) rate: $\min_{1\leq k\leq K}|f’(x^k)| \leq \sqrt{\frac{2L_0(f(x^0)-f^\star)}{(1-2\beta)(K+1)}} + \frac{L_1(f(x^0)-f^\star)}{(1-2\beta)(K+1)}$. For $\beta=0$, this is a better rate than the best known one for $(L_0, L_1)$-smoothness [4, Theorem 3.1].
> > >
> > > We can thus see that anisotropic smoothness is not only more general than $(L_0, L_1)$-smoothness (see also our reply to reviewer VEKm) but also leads to faster algorithms.
> > >
> > >
> > > ---
> > >
> > > **References**
> > >
> > > [4] Vankov, Daniil, et al. "Optimizing $(L_0, L_1) $-Smooth Functions by Gradient Methods." arXiv preprint arXiv:2410.10800 (2024).
> > >
> > > [5] Lobanov, Aleksandr, et al. "Linear Convergence Rate in Convex Setup is Possible! Gradient Descent Method Variants under $(L_0, L_1) $-Smoothness." arXiv preprint arXiv:2412.17050 (2024).
> > >
> > > [6] Chen, Ziyi, et al. "Generalized-smooth nonconvex optimization is as efficient as smooth nonconvex optimization." ICML (2023).
> > >
> > > [7] Zhang, Jingzhao, et al. "Why gradient clipping accelerates training: A theoretical justification for adaptivity." arXiv preprint arXiv:1905.11881 (2019).

---

> > > > ### Comment · Reviewer_tUyH · 2025-08-06
> > > >
> > > > Thanks for your response! It has addressed my concerns, so I have raised my rating.

---

### Official Review · Reviewer_QeMc · 2025-06-29

**Clarity:** 4
**Significance:** 3
**Originality:** 4
**Rating:** 5
**Confidence:** 2

**Summary:**

This paper investigates smooth noncovnex optimization problem using anisotropic gradient descent (GD) method & its extensions. They combine the notion of momentum with nonlinear preconditioned GD & propose a Heavy-Ball (HB)-type method with a convergence rate of $\mathcal{O}(L/K)$ for a strongly convex reference function $\phi$ & $f$ being anisotropically smooth relative to $\phi$ (a notion that was proposed by previous works). Later, they extend their convergence result to the scenario when $f$ satisfies a generalized PL inequality and $\phi$ is 2-subhomogenous. Additionally, they study their method for functions that satisfy a preconditioned-smoothness assumption which contains $(L_0-L_1)$-smoothness as a special case. A convergence analysis of the stochastic preconditioned GD method under a more general noise assumptions revealed that the convergence is not guaranteed without variance reduction techniques (like increasing batch sizes). Numerical experiments revealed competetive behaviour of the proposed method.

**Questions:**

First, thank you for your efforts.

1- In line 117 it is mentioned that this work mainly focuses on strongly convex reference functions. Then they propose some examples. What are other examples of strongly convex reference functions?

2- Did you ensure that all the methods in your simulations are compaired fairly in terms of the choice of the step-sizes? I'm mainly referring to the neural network simulations where the step-size was chosen through sweeping on $\{1,0.1,0.01\}$.

3- Why ADAM was not compared in the training on CIFAR10 ResNet18 experiment with momentum?

4- Do you think it would be possible to extend these results to the constrained optimization?

Thanks!

**Ethical Concerns:**

["NO or VERY MINOR ethics concerns only"]

**Final Justification:**

After reading all the comments, I decide to raise my score to 5 as I see clear positive responses from other reviewers who responsed. The authors have clearly addressed my concerns regarding their simulation results. Due to comparative novelty aspects existing in NeurIPS, I do not raise my score to 6 as this work is developing on the same line of prior works like reference [30]. However, still they bring valuable results as mentioned in the "Strengths" section of my initial comment.

**Limitations:**

Yes

**Paper Formatting Concerns:**

No formatting issues

**Quality:**

3

**Strengths And Weaknesses:**

**Strengths**

1- The problem is well-motivated. Although, the gradient clipping method is more explored and studied in the literature, its general form (1) is less attended. This is the general motivation in this work. (Lines 42-54)

2- The contribution is good. To further study the general form of nonlinear preconditioned GD, they propose an HB-type method with its convergence analysis under the assumptions of anisotropic smooth relative to $\phi$ (lines 155-156 theorem 2.1), a generalized PL inequality where $\phi$ is 2-subhomogenous (lines 197-201 theorem 2.3), and a preconditioned Lipschitz continuity (lines 225-228 theorem 2.6) . A stochastic extension is also studied (for the first time as the authors claim) under a slightly more general noise assumption (lines 241-243 theorem 3.1). Additionally, they show that if the mini-batch size increase (classical approach to reduce the variance estimation error), the true convergence is possible in theorem 3.4.


**Weaknesses**

1- The writing could improve, sometimes the reader might feel lost or diconnected. Adding a bit more coherency might help with this (this point exists throughout the whole paper. Examples are Paragraph on line 22 is incoherent with the previous paragraph, paragraph on line 203 hard to read sentence on line 207).

2- ADAM is a momentum-based algorithm, however it was not compared in the Numerical simulations (Figure 1, right).

---

> ### Author Rebuttal · Authors · 2025-07-30
>
> We thank the reviewer for their positive evaluation of our paper, and we appreciate their recognition of the significance of our work.
>
> In the following we address the **Weaknesses** stated by the reviewer.
>
> > 1\. The writing could improve, sometimes the reader might feel lost or diconnected. Adding a bit more coherency might help with this (this point exists throughout the whole paper. Examples are Paragraph on line 22 is incoherent with the previous paragraph, paragraph on line 203 hard to read sentence on line 207).
>
> We acknowledge that our writing could improve at places in the manuscript. We will refine the writing by simplifying the language and providing additional details.
>
> > 2\. ADAM is a momentum-based algorithm, however it was not compared in the Numerical simulations (Figure 1, right).
>
> We did not include Adam in Figure 1 (right) because this experiment is the same (Resnet18 on the Cifar10 dataset) as the one shown in Figure 1 (center), where Adam is already included. We aimed to avoid overcrowding the image on the right.
>
> In what follows we answer the **Questions** asked by the reviewer.
>
> > 1\. In line 117 it is mentioned that this work mainly focuses on strongly convex reference functions. Then they propose some examples. What are other examples of strongly convex reference functions?
>
> There are many interesting strongly convex reference functions. In order to avoid repetition we only present the 1d kernel functions $h$ that generate $\phi$, along with the corresponding preconditioner ${h^*}'$, in the following table:
> | $h(x)$                                              | ${h^*}'(y)$                        |
> |-----------------------------------------------------|------------------------------------|
> | $\exp(\|x\|)-\|x\|-1$                               | $\ln(1+\|y\|)\text{sgn}(y)$        |
> | $1-\sqrt{1-x^2}$                                    | $\frac{y}{\sqrt{1+y^2}}$           |
> | $\frac{1}{2}x^2 + \delta_{[-1,1]}(x)$               | $\text{min}(1, \text{max}(-1, y))$ |
> | $x \text{arctanh(x)}-\ln(\cosh(\text{arctanh(x)}))$ | $\tanh(y)$                         |
> | $-\frac{2}{\pi}\log(\cos(\frac{\pi x}{2}))$                 | $\frac{2}{\pi}\text{arctan(y)}$    |
>
> We remark that the second and third reference functions lead to a simplified version of Adagrad and the gradient clipping algorithm respectively through [1, Examples 1.5 & 1.7].
>
> > 2\. Did you ensure that all the methods in your simulations are compaired fairly in terms of the choice of the step-sizes? I'm mainly referring to the neural network simulations where the step-size was chosen through sweeping on $1, 0.1, 0.01$.
>
> In order to clear possible confusion we remark that except for the experiment with momentum, in all the other ΝΝ experiments we tuned the methods via an adaptive gridsearch. We did not use the same gridsearch for the experiment with momentum to lower the computational cost, since SGDm achieves state-of-the-art performance with these parameters.
>
> > 3\. Why ADAM was not compared in the training on CIFAR10 ResNet18 experiment with momentum?
>
> Please see our comment on the second weakness.
>
> > 4\. Do you think it would be possible to extend these results to the constrained optimization?
>
> Indeed, we believe that an extension to the constrained case is certainly possible and is interesting future work. We remark that another advantage of the anisotropic smoothness framework that we study is that it has a natural proximal extension for additive nonsmooth problems [2].
>
> ---
> **References:**
>
> [1] Oikonomidis, Konstantinos, et al. "Nonlinearly Preconditioned Gradient Methods under Generalized Smoothness." ICML (2025).
>
> [2] Laude, Emanuel, and Panagiotis Patrinos. "Anisotropic proximal gradient." Mathematical Programming (2025): 1-45.

---

> > ### Comment · Reviewer_QeMc · 2025-08-02
> >
> > I thank the authors for their detailed explanation and clarifications.
> >
> > Just one more clarification: The caption of Figure 1 states that the simulation in the middle is for methods without using momentum and it includes Adam, (I suppose you discarded the momentum parameter in your implementation), then the figure on the right is for the methods with momentum, but it does not include Adam. Do you mean that momentum does not play a crucial role for the Adam? Or the figure in the middle is Adam with momentum?

---

> > > ### Author Response · Authors · 2025-08-02
> > >
> > > Dear reviewer QeMc
> > >
> > > The middle figure includes Adam with the standard momentum parameters $\beta_1=0.9, \beta_2=0.999$. See line 34 of the file `neural_networks/main.py` in the supplementary material:
> > > ```
> > > return problem(get_device(), lambda params: torch.optim.Adam(params, lr)).train(epochs=epochs)
> > > ```
> > > To avoid confusion, we will move Adam to the rightmost figure.
> > >
> > > We apologize for the misunderstanding.

---

> > > > ### Comment · Reviewer_QeMc · 2025-08-04
> > > >
> > > > Thank you for your response.
> > > >
> > > > Yes, I believe that would be improving the presentation there. I might raise my score in the next phase.

---

### Official Review · Reviewer_28hb · 2025-06-29

**Clarity:** 3
**Significance:** 2
**Originality:** 2
**Rating:** 4
**Confidence:** 3

**Summary:**

This paper presents a theoretically rigorous and practically motivated study of nonlinearly preconditioned gradient methods (NPGMs), specifically extending them with momentum (m-NPGM) and developing a stochastic variant. The paper analyzes convergence under anisotropic smoothness, a generalization of Lipschitz smoothness, and establishes novel results under generalized PL conditions and (L0, L1)-smoothness. It also provides empirical comparisons on deep learning and matrix factorization tasks.

**Questions:**

see above

**Ethical Concerns:**

["NO or VERY MINOR ethics concerns only"]

**Final Justification:**

I want to thank the authors for providing the response. I am satisfied with the response and have raised the rating to 4.

**Limitations:**

see above

**Quality:**

2

**Strengths And Weaknesses:**

Strength: 1) Theory: This work introduces a novel heavy-ball extension of nonlinear preconditioning methods with provable convergence guarantees in the general nonconvex setting under a generalized PL condition. This is the first result regarding the convergence of the method with momentum under anisotropic smoothness conditions.
2) Assumption and setting: the analysis is established under anisotropic smoothness and preconditioned Lipschitz conditions, which cover many applications. In particular, the analysis is a non-trivial extension of the previous work due to the particular structure of the anisotropic descent inequality and the lack of global upper bounds for the gradient difference term. Therefore, the techniques developed here have the potential to be applied to develop and analyze other algorithms in this setting.

Weakness: The presentation of the content lacks details. All the main results are presented right after the assumptions/conditions, without any elaboration on the proof sketch and technical novelty. The paper discusses a variety of settings, but could focus on one or two of them to further elaborate the technical details. The experimental section could be strengthened with ablation on choice of reference function and effect of momentum coefficient on convergence, etc.

---

> ### Author Rebuttal · Authors · 2025-07-30
>
> We appreciate the reviewer’s time and effort in providing feedback.
>
> In the following we address the **Weaknesses** stated by the reviewer.
>
> > The presentation of the content lacks details. All the main results are presented right after the assumptions/conditions, without any elaboration on the proof sketch and technical novelty. The paper discusses a variety of settings, but could focus on one or two of them to further elaborate the technical details.
>
> We appreciate any feedback on the presentation of our work. However, in this case the comment feels unjustified, as we motivate below.
>
> One of our main theorems is Theorem 2.1 and the three paragraphs that follow it are dedicated to discussing both the difficulties of the setting that we study and its limitations. Then, we present the generalized PL condition and briefly explain it in one paragraph, but we provide additional details on it and further study its relation to the classical PL condition in Appendix D. We do not discuss the difficulty of the setting after Theorem 2.3, since this is similar to Theorem 2.1.
>
> We then move on to establishing the preconditioned Lipschitz continuity assumption, which we state after an introductory paragraph. Following the statement of the assumption we show that it holds for Lipschitz smooth functions and even for $(L_0,L_1)$-smooth ones, thus showcasing its generality and applicablity. Then, we state our convergence theorem under this new assumption. We do not provide a discussion after this statement due to spacing constraints, but our proofs are presented in full detail in the appendix.
>
> Regarding Section 3, we present it in a way such that the main convergence result Theorem 3.4 follows naturally. The logic of the presentation is as follows: we first show in Theorem 3.1 the convergence of the method up to $\sigma^2$ under a new noise assumption. Then, in Proposition 3.2 we show that this noise assumption is actually implied by the standard bounded variance assumption and in Example 3.3 we provide an example where the bounded variance assumption fails and our noise assumption holds. Following that we provide the convergence result Theorem 3.4 where we assume bounded variance and that we have access to a minibatch oracle. Finally, we discuss the limitations and advantages of our approach.
>
> In general, we agree that in some places of the manuscript we could provide more details and the page limit constraint is the main reason we have not done so. Nevertheless, we do not believe that this is a valid reason to reject the paper.
>
> > The experimental section could be strengthened with ablation on choice of reference function and effect of momentum coefficient on convergence, etc.
>
> Regarding the choice of preconditioner in our experiments, please see our comment on weakness 3 stated by reviewer A43W. As for the ablation study on the momentum parameter, we consider this an interesting research direction, as the current paper primarily focuses on theoretical analysis.
>
> We believe that we have addressed all the reviewer's concerns and weaknesses in our paper, and we kindly ask the reviewer to reevaluate their rating.

---

### Official Review · Reviewer_VEKm · 2025-06-30

**Clarity:** 3
**Significance:** 3
**Originality:** 3
**Rating:** 4
**Confidence:** 3

**Summary:**

This paper proposes a nonlinearly preconditioned heavy-ball-type algorithm and provides its convergence guarantees in the general nonconvex setting and under a generalized PL condition. It also presents a stochastic variant and analyzes its convergence behavior under both new and standard noise assumptions. Finally, it tests the proposed methods and shows their good performance on various tasks, including neural network training and matrix factorization.

**Questions:**

1. It is better to explain sigmoid preconditioners in the main text.
2. Is there any relationship between the so-called Hyperbolic Gradient Descent with the gradient descent in hyperbolic space [1]?
3. Is it better to use $\cosh(\cdot)-1$ instead of $\cosh-1$?
4. To obtain the next iterate, Equation (2) is minimized over what?
5. Define $\delta_{\{0\}}(x)$ in Line 103
6. Define $\text{rge}$ in Equation (3)
7. It is better to give the formulation of the stochastic optimization problem at the beginning of Section 3.
8. What about the numerical performances of algorithms induced by other reference functions?
9. Is it possible to include numerical experiments on PL-type functions?
---
[1] Wilson, B., & Leimeister, M. (2018). Gradient descent in hyperbolic space. arXiv preprint arXiv:1805.08207.

**Ethical Concerns:**

["NO or VERY MINOR ethics concerns only"]

**Final Justification:**

I believe the authors have addressed all my concerns. They are also committed to incorporating several discussions into the revised manuscript.

**Limitations:**

Yes.

**Paper Formatting Concerns:**

I have not found any major formatting issues in this paper.

**Quality:**

3

**Strengths And Weaknesses:**

**Strengths**

1. This paper extends the nonlinearly preconditioned gradient descent method by incorporating a momentum term and studies the convergence behavior under different conditions.

2. This paper also proposes a stochastic variant of the base method and analyzes it under different noise assumptions.

3. The numerical experiments consider two different applications and show the good performance of the proposed methods.

**Weaknesses**
1. Some concepts are used without any explanations, such as $\Phi$-convexity. It may be better to give a brief introduction in the appendix.

2. It is better to compare the standard convergence analysis of gradient clipping under the $(L_0, L_1)$-smooth condition and the one under this anisotropic smoothness. For example, does the paper obtain a stronger result in the stochastic setting? Does the analysis become simpler? Are the assumptions weaker? I think the current manuscript lacks a discussion on this aspect.

---

> ### Author Rebuttal · Authors · 2025-07-31
>
> We appreciate the reviewer’s time and effort in providing feedback and their acknowledgement of our papers' novelty, quality and clarity.
>
> In the following we address the **Weaknesses** stated by the reviewer.
> > 1\. Some concepts are used without any explanations, such as $\Phi$-convexity. It may be better to give a brief introduction in the appendix.
>
> We would like to thank the reviewer for their suggestion. Indeed, we will provide additional details on the important concept of $\Phi$-convexity in a revised version of the manuscript.
>
> > 2\. It is better to compare the standard convergence analysis of gradient clipping under the $(L_0, L_1)$-smooth condition and the one under this anisotropic smoothness. For example, does the paper obtain a stronger result in the stochastic setting? Does the analysis become simpler? Are the assumptions weaker? I think the current manuscript lacks a discussion on this aspect.
>
> We would like to stress that although there exist connections between the two notions of smoothness, they are not equivalent and thus directly comparing the analysis of the methods is not straightforward.
>
> > Are the assumptions weaker?
>
> From [1, Proposition 2.9] we know that anisotropic smoothness encompasses a class of functions that is wider than that of $(L_0, L_1)$-smooth functions. In other words, anisotropic smoothness is a **weaker assumption** than $(L_0, L_1)$-smoothness. To better see this, note that for $\phi(x)= -L_0/L_1 (-\\|x\\|-\ln(1-\\|x\\|))$, the second-order sufficient condition for anisotropic smoothness from [1, Definition 2.4] takes the following form:
> $$
> \nabla^2 f(x) \prec ( L_0 +  L_1\\|\nabla f(x)\\|)I + \left( L_1 + \frac{ L_1^2}{L_0} \\|\nabla f(x)\\|\right) \frac{\nabla f(x)\nabla f(x)^\top}{\\|\nabla f(x)\\|},
> $$
> while for $(L_0, L_1)$-smoothness it is $\\|\nabla^2 f(x)\\|\leq L_0 + L_1\\|\nabla f(x)\\|$.
> Therefore, we can see that this second-order sufficient condition 1) does not require lower bounds on the eigenvalues of $\nabla^2 f(x)$ and 2) can be less restrictive than the one for $(L_0, L_1)$-smoothness stated above. Some simple examples of functions that are anisotropically smooth but not $(L_0, L_1)$ include $1-\sqrt{1-x^2}$, $\exp(x^2)$, $-|x|-\ln(1-|x|)$ (by choosing as $\phi$ the function itself). It is also important to note that our preconditioned Lipschitz continuity Assumption 2.4 is at least as general as $(L_0, L_1)$-smoothness and can be even less restrictive when the Jacobian of $\nabla \phi^* \circ \nabla f$ is symmetric, following the discussion on lines 204-212 of the manuscript.
>
> We remark that another advantage of the anisotropic smoothness framework is that it can better describe the functions that are between Lipschitz smoothness and $(L_0, L_1)$-smoothness in that it generates tighter upper bounds. For further details please see our reply to reviewer tUyH.
>
> > Does the analysis become simpler?
>
> Our analysis covers a whole family of algorithms (parameterized by the reference function $\phi$) under a very general condition and can thus be considered simpler, since it does not require further effort for every specific instance of the algorithm and the corresponding descent inequality, as is standard in the related literature. Moreover, it is illuminating in that it links the choice of the preconditioner (and thus algorithm) with the function class of the cost function.
>
> > Does the paper obtain a stronger result in the stochastic setting?
>
> In the stochastic setting we obtain standard convergence guarantees for this type of methods (without some variance reduction technique) [3, Section 4.1], but under a different condition.
>
> We aim to add a discussion on this subject in a revised version of the manuscript.
>
> In what follows we answer the **Questions** asked by the reviewer.
>
> 1. We thank the reviewer for their suggestion, we will include some discussion on the sigmoid shape of the preconditioners in the main text.
> 2. We are not aware of any relationship between HGD and gradient descent in hyperbolic space. However, the general update described in (1) is connected to works from the geometry of optimal transport literature [4, 5].
> 3. Thank you for the suggestion, we will replace $\cosh-1$ with $\cosh(\cdot)-1$.
> 4. The main iterate is generated by minimizing the r.h.s. of the inequality (2) over $x$. We will specify this in the manuscript.
> 5. We will include the suggested changes in a revision.
> 6. We will include the suggested changes in a revision.
> 7. We will include the suggested changes in a revision.
> 8. In our paper we focus mostly on the algorithms generated by $h(x) =\cosh(x)-1$, since their behavior is between that of plain GD and gradient clipping (see also the response to reviewer A43W). We did not provide experiments with reference functions that have bounded domains, since they perform some form of gradient clipping for which there already exists a rich literature.
> 9. We believe that in order for such experiments to be meaningful, the algorithms should be applied with the correct stepsize obtained by the theory. For that reason we have left such a project for future work. Nevertheless a toy experiment is presented in our reply to reviewer tUyH, where the function satisfies the generalized PL inequality along the sequence of iterates.
>
> We believe that we have addressed all the reviewer's concerns and weaknesses, and we kindly ask the reviewer to reevaluate their rating.
>
> ---
> **References:**
>
> [1] Oikonomidis, Konstantinos, et al. "Nonlinearly Preconditioned Gradient Methods under Generalized Smoothness." ICML (2025).
>
> [2] Laude, Emanuel, and Panagiotis Patrinos. "Anisotropic proximal gradient." Mathematical Programming (2025): 1-45.
>
> [3] Lu, Haihao, Robert M. Freund, and Yurii Nesterov. "Relatively smooth convex optimization by first-order methods, and applications." SIAM Journal on Optimization 28.1 (2018): 333-354.
>
> [4] Gangbo, Wilfrid, and Robert J. McCann. "The geometry of optimal transportation." (1996): 113-161.
>
> [5] Figalli, Alessio, Young-Heon Kim, and Robert J. McCann. "When is multidimensional screening a convex program?." Journal of Economic Theory 146.2 (2011): 454-478.

---

### Official Review · Reviewer_A43W · 2025-07-02

**Clarity:** 3
**Significance:** 3
**Originality:** 3
**Rating:** 4
**Confidence:** 3

**Summary:**

The paper proposes nonlinearly pre-conditioned gradient methods (NPGMs) that combine the dual-space/anisotropic-descent framework with heavy-ball momentum and with a stochastic variant. It establishes global convergence bounds for the momentum version under anisotropic smoothness and shows a linear rate when a generalized PL condition holds. For the stochastic setting, the authors derive non-asymptotic rates under a novel noise assumption that subsumes bounded variance and prove linear convergence under generalized PL. Finally, they include experiments on MNIST/CIFAR-10 classification and matrix-factorization tasks.

**Questions:**

See weaknesses

**Ethical Concerns:**

["NO or VERY MINOR ethics concerns only"]

**Final Justification:**

The authors have made an adequate job addressing my concerns. Therefore I maintain my initial rating of 4.

**Limitations:**

Yes

**Quality:**

3

**Strengths And Weaknesses:**

**Strengths**
1. The paper is well structured and has a nice flow.
2. They provide rigorous proofs for all stated theorems, and the assumptions are clearly presented. In particular, they both deterministic and stochastic analyses, including mini-batch dependence.
3. The authors give the first convergence proof for heavy-ball momentum inside anisotropic preconditioning setting.

**Weaknesses**
1.  In the statement of Theorem 2.1 the authors make the restriction $\beta<0.5$, limiting practical momentum choices (usually $\beta=0.9$), however, they note this as an open question.
2. The stochastic results (Section 3) cover only the base (no-momentum) method. Following the work from Section 2, one would expect results for the momentum-based method. After all, this is what practitioners usually deploy.
3. Empirical accuracy gains over tuned SGD/Adam are modest on CIFAR-10; error bars overlap after 200 epochs.

---

> ### Author Rebuttal · Authors · 2025-07-30
>
> We thank the reviewer for their positive evaluation of our paper.
>
> In the following we address the **Weaknesses** stated by the reviewer.
>
> > 1\. In the statement of Theorem 2.1 the authors make the restriction $\beta \in [0, 0.5)$, limiting practical momentum choices (usually $\beta = 0.9$), however, they note this as an open question.
>
> Indeed, Theorem 2.1 covers momentum parameters $\beta \in [0,0.5)$, a limitation that we acknowledge in our manuscript. Nevertheless, in Theorem 2.6 we provide convergence guarantees of the method for $\beta \in [0,1)$ under a condition that we show is more general than standard Lipschitz smoothness and is at least as general as $(L_0, L_1)$-smoothness. This condition can also be less restrictive than $(L_0, L_1)$-smoothness, when the Jacobian of $\nabla \phi^* \circ \nabla f$ is symmetric, following the discussion in lines 204-212. In that sense our result Theorem 2.6 is state-of-the-art, especially considering that it covers the whole family of algorithms described in Algorithm 1 and it does not require considering specific instances (e.g. normalized gradient) as is standard in the literature. Therefore, Theorem 2.1 describes an even broader result, since it holds under an even less restrictive condition, with the caveat that $\beta \in [0, 0.5)$.
>
> > 2\. The stochastic results (Section 3) cover only the base (no-momentum) method. Following the work from Section 2, one would expect results for the momentum-based method. After all, this is what practitioners usually deploy.
>
> In this paper we have indeed not studied the proposed algorithm with momentum in the stochastic setting. We believe that this is an important research direction that deserves separate work, especially when considering that our paper is the first that studies the momentum extension and the stochastic version of the base method in the general nonlinearly preconditioned gradient framework.
>
> > 3\. Empirical accuracy gains over tuned SGD/Adam are modest on CIFAR-10; error bars overlap after 200 epochs.
>
> In our experiments we have focused on the algorithms generated by $\phi(x) = \cosh(\|x\|)-1$ and $\phi(x)=\sum_{i=1}^n \cosh(x_i)-1$ (called HGD) which we find interesting because they behave in a way that is between standard GD and gradient clipping. This intermediate behavior is especially clear in one dimension, where it can be illustrated by plotting the preconditioner $\sinh^{-1}(y)$. In that sense, the aim of our experimental section is to show that it keeps the best of both worlds in that it performs similarly to or even better than state-of-the-art methods in a variety of problems including NN training, while at the same time having stronger theoretical guarantees than GD. Therefore, HGD is a robust solver that can tackle a variety of problems, supported by theoretical guarantees that extend beyond the usual Lipschitz smoothness assumptions.

---

> > ### Comment · Reviewer_A43W · 2025-08-05
> >
> > I thank the authors for their response. I will maintain my positive score.

---

### Note · Authors · 2025-08-13

We thank the reviewers for their time and effort in providing feedback.

We would like to highlight some important aspects and contributions of our work:
- We introduce a family of heavy ball-type algorithms and analyze their convergence under a condition that is both less restrictive than $(L_0, L_1)$-smoothness and tighter (Theorems 2.1 & 2.6). We note that our unifying framework is supported by a simplified and illuminating analysis that not only covers existing algorithms but also naturally leads to new ones.
- For the same family of methods we also prove their linear convergence under a generalized PL inequality (Theorem 2.3) for which we provide examples and connections to the standard PL inequality.
- We perform the first stochastic analysis of the nonlinearly preconditioned gradient method, showing convergence up to $\sigma^2$ under a noise assumption that is less restrictive than bounded variance (Theorem 3.1) and obtaining standard convergence guarantees under bounded variance (Theorem 3.4).

After the rebuttal, we have incorporated the following changes into the manuscript:
- We have added further details on important concepts such as $\Phi$-convexity, the generalized PL inequality and the preconditioned Lipschitz continuity condition that we introduce (Assumption 2.4). Moreover, we have provided additional explanations of the main steps in our proofs and simplified parts of the text that were too convoluted, thereby enhancing the readability of our manuscript.
- We have included a discussion on the advantages of anisotropic smoothness compared to other notions of generalized smoothness, highlighting that it is more general than $(L_0, L_1)$-smoothness (incorporating our reply to reviewer VEKm) and that it leads to tighter descent inequalities (incorporating our reply to reviewer tUyH).

---

### Decision · Program_Chairs · 2025-09-17

**Decision:**

Accept (poster)

**Comment:**

This paper studies preconditioned gradient methods for smooth nonconvex optimization problems. This work is a follow up to the work of Oikonomidis, Quan, Laude, and Patrinos [30] who introduces the idea of preconditioned gradient methods which generalize gradient clipping techniques. Moreover, the assumptions which they prove convergence generalizes (L_0,L_1)-smooth functions.

In this paper, they analyze (i) a heavy ball variant of the method (algorithm 1) and (ii) a stochastic version of the method (which as theorem 3.4 specifies requires very large minibatches such that the gradient noise is less than 1/K where K is the number of iterations). Their numerical results are okay with no meaningful gains training CIFAR10/MNIST (figure 1) but significant gains demonstrated on matrix factorization with the Movielens 100K dataset (figure 2). One weakness of these empirical results is that there is limited modern relevance of gains on this task.

Comments:

Please explain clearly how heavy tail distributions are captured by your assumption that E[\phi(\grad \phi*(\grad f(x)))] <= \sigma^2.

“We set the learning rate by performing a parameter sweep over {1, 0.1, 0.01}.” Use a finer grid.